# Online Selective Conformal Inference: Errors and Solutions

**Yusuf Sale**                                                                *yusuf.sale@ifi.lmu.de*
*Institute of Computer Science*
*Ludwig-Maximilians Universität München*
*Munich Center for Machine Learning (MCML)*

**Aaditya Ramdas**                                                          *aramdas@stat.cmu.edu*
*Departments of Statistics and Machine Learning*
*Carnegie Mellon University*

**Reviewed on OpenReview:** *https: // openreview. net/ pdf? id=PjIQwFyPO7*

## Abstract

In online selective conformal inference, data arrives sequentially, and prediction intervals are constructed only when an online selection rule is met. Since online selections may break the exchangeability between the selected test datum and the rest of the data, one must correct for this by suitably selecting the calibration data. In this paper, we evaluate existing calibration selection strategies and pinpoint some fundamental errors in the associated claims that guarantee selection-conditional coverage and control of the false coverage rate (FCR). To address these shortcomings, we propose novel calibration selection strategies that provably preserve the exchangeability of the calibration data and the selected test datum. Consequently, we demonstrate that online selective conformal inference with these strategies guarantees both selection-conditional coverage and FCR control. Our theoretical findings are supported by experimental evidence examining trade-offs between valid methods.

## 1 Introduction

*Conformal prediction* (Vovk et al., 2005) has gained significant traction in recent years as a method for uncertainty quantification equipped with statistical guarantees. Conformal prediction provides prediction intervals (or sets) that achieve a predefined coverage level, regardless of the underlying data distribution. In contrast to the *full* (or transductive) conformal procedure, the computationally more efficient *split* (or inductive) conformal prediction (Papadopoulos et al., 2002; Lei et al., 2018) splits the available data into two parts: one for model training and another for calibration. More formally, suppose a pre-trained model $\hat{\mu} : \mathcal{X} \rightarrow \mathcal{Y}$ is given, mapping features $X \in \mathcal{X}$ to predictions of the label $Y \in \mathcal{Y}$. Further, denote by $\{(X_i, Y_i)\}_{i=1}^{n}$ an independently labeled *calibration* data set. Given the current feature $X_{n+1}$, the goal is to construct a prediction interval for the unseen label $Y_{n+1}$. If the data sequence $\{(X_i, Y_i)\}_{i=1}^{n+1}$ is exchangeable, the constructed prediction interval $\widehat{C}_n(X_{n+1})$ contains $Y_{n+1}$ with high probability (*viz.* the prediction interval $\widehat{C}_n(X_{n+1})$ is *valid*). Specifically, we have

$$\mathbb{P}\{Y_{n+1} \in \widehat{C}_n(X_{n+1})\} \geq 1 - \alpha, \tag{1}$$

where the probability is taken over $\{(X_i, Y_i)\}_{i=1}^{n+1}$, emphasizing the *marginal* nature of (1).

Conformal prediction has also been successfully adapted for *online* applications. Assume that the data sequence $\{(X_t, Y_t)\}_{t \geq 0} \subseteq \mathcal{X} \times \mathcal{Y}$ is observed *sequentially*: at each time $t$, we observe the previous label $Y_{t-1}$ and the current feature vector $X_t$. While much of the literature has focused on addressing cases where exchangeability does not hold—proposing generalizations or relaxations of this assumption (Tibshirani et al.,

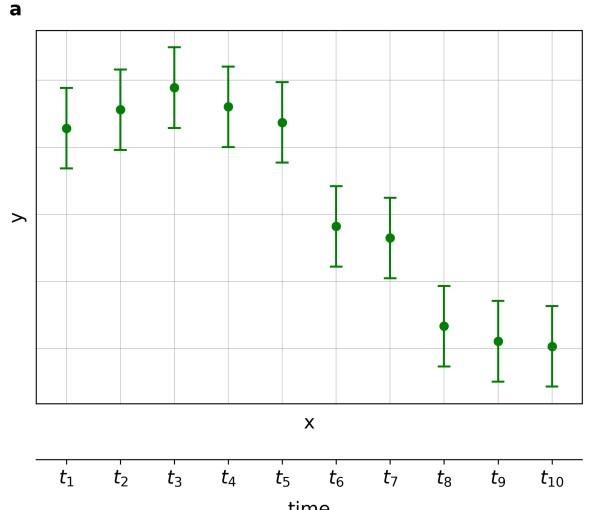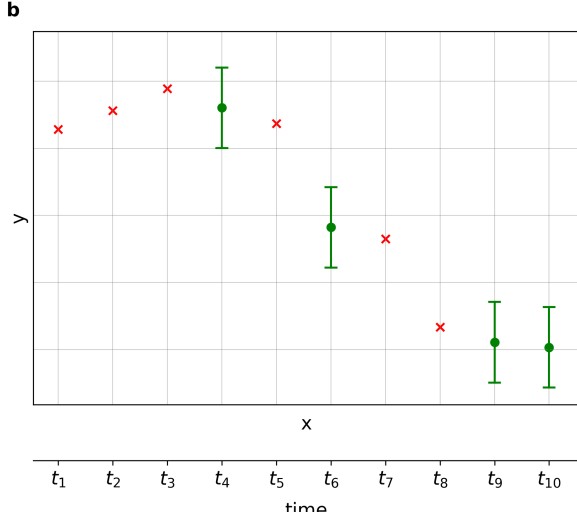

Figure 1: An illustrative example of the online selective (conformal prediction) setting. **(a)** Usual online conformal prediction setting, where one constructs for *each* time $t \geq 0$ a prediction interval. **(b)** In the online *selective* conformal predictive setting, we only report prediction intervals for selected ($\bullet$) times, while no prediction intervals are constructed for those that are not selected ($\times$).

2019; Gibbs & Candes, 2021; Barber et al., 2023)—our focus is different: for us the data is exchangeable, but challenges (indeed, deviations from exchangeability) are caused by selective querying/reporting. To elaborate, suppose data arrive sequentially, and we wish to *selectively* report prediction intervals (see Figure 1 for an illustration). The selection process (formally defined in Section 2) might, for instance, dictate that a prediction interval for $Y_t$ is reported only if $X_t > 2$. More generally, the selection rule may depend on past decisions (in a restricted way formalized later)—for example, whether a prediction interval was reported for previous observations. Consequently, prediction intervals are not constructed for every observation, adding a layer of selective decision-making to online conformal inference, referred to as *online selective conformal inference.*

In the online selective setting, it is natural to require that prediction intervals are valid conditioned on the selection event—a metric called *selection-conditional coverage*:

$$\forall t \geq 0 \, : \, \mathbb{P}\{Y_t \in \widehat{C}_t(X_t) \,|\, S_t = 1\} \geq 1 - \alpha. \tag{2}$$

Intuitively, a prediction interval should be valid whenever a selection is made, that is, whenever $S_t = 1$. Since the data itself is exchangeable, this requirement might seem innocuous at first glance. However, it is important to highlight a critical aspect. The (potential) dependence between online selections and the prediction intervals can give rise to temporal *multiplicity.* In the statistical literature, this issue was previously recognized in the context of constructing confidence sets. It was first noted in the offline setting by Benjamini & Yekutieli (2005) and later addressed in the online regime by Weinstein & Ramdas (2020). Recently, Bao et al. (2024a) addressed the same problem and proposed a procedure called *calibration after adaptive pick* (CAP) to construct conformal prediction intervals that satisfy (2). However, despite the claims of Bao et al. (2024a), we demonstrate that the proposed method does *not*, in fact, guarantee selection-conditional coverage.

Other metrics to assess the errors associated with the reported prediction intervals are the false coverage proportion (FCP) and false coverage rate (FCR) (Weinstein & Ramdas, 2020)

$$\text{FCP}(T) = \frac{\sum_{t=0}^{T} S_t \mathbb{1}\{Y_t \notin \widehat{C}_t(X_t)\}}{1 \vee \sum_{t=0}^{T} S_t}, \quad \text{FCR}(T) = \mathbb{E}\left[\frac{\sum_{t=0}^{T} S_t \mathbb{1}\{Y_t \notin \widehat{C}_t(X_t)\}}{1 \vee \sum_{t=0}^{T} S_t}\right], \tag{3}$$

where $a \vee b = \max(a, b)$ for $a, b \in \mathbb{R}$. The FCP measures the proportion of selected instances where the true label $Y_t$ falls outside the prediction interval $\widehat{C}_t(X_t)$, while the FCR is the expected value of the FCP.

Weinstein & Ramdas (2020) proposed a method called LORD-CI, which effectively controls the FCR below a target level $\alpha$. Although originally proposed for confidence sets, the authors note that the LORD-CI procedure can be adapted to the online conformal prediction framework. We find that their claim is true in a particular restricted setting, but that a broader extension is *not* straightforward and highlight the difficulties involved.

The key challenge in the online selective conformal inference setting is the selection of calibration data to form $\widehat{C}_t$ when $S_t = 1$. Specifically, since the selected data point might not be exchangeable with all preceding data (due to the selection), how should calibration data be chosen so as to *restore* the exchangeability of the test datum and the calibration data? To address this, we discuss various calibration selection *strategies* within the framework of online selective conformal inference, evaluating their ability to ensure selection-conditional coverage. Our theoretical results reveal that many existing calibration selection strategies fail to meet this quite stringent criterion. To bridge this gap, we propose novel calibration selection strategies with theoretical guarantees for selection-conditional coverage, supported by simulation studies. Beyond selection-conditional coverage, we further investigate whether the (seemingly) weaker metric of FCR can be successfully controlled by online selective conformal prediction algorithms instantiated with both existing and novel calibration selection strategies. Interestingly, we find that no existing strategy achieves this, while our proposed methods provide provable guarantees.

The remainder of the paper is organized as follows. Section 2 formally defines the problem and introduces necessary notation. In Section 3, we outline the online selective conformal inference procedure and discuss several calibration selection strategies, including both existing and novel approaches. Section 4 assesses, empirically and theoretically, whether existing calibration selection strategies ensure selection-conditional coverage and provides theoretical guarantees for our proposed methods. Section 5 investigates FCR control, followed by a discussion of other related conformal methods. We conclude with a summary of related work and a brief discussion. Omitted proofs are deferred to Appendix B.

## 2 Problem Setup

Consider data $\{(X_t, Y_t)\}_{t \geq 0} \subseteq \mathcal{X} \times \mathcal{Y}$ arriving sequentially: at time $t$, we observe the previous label $Y_{t-1}$ (if $t > 0$) and the new feature $X_t$. For brevity, we also write $Z = (X, Y)$ and $\mathcal{Z} = \mathcal{X} \times \mathcal{Y}$. We assume that a pre-trained model $\hat{\mu} : \mathcal{X} \to \mathcal{Y}$ is given, where $\widehat{Y}_t = \widehat{\mu}(X_t)$ denotes a prediction of the label $Y_t$. Let $\mathcal{S}_t : \mathcal{X} \to \{0, 1\}$ be an online selection *rule*, and correspondingly denote by $S_t = \mathcal{S}_t(X_t)$ the selection *indicator*. Here, $S_t = 1$ indicates that a prediction interval for $Y_t$ will be reported. Further, we denote by $\mathcal{F}_{t-1} = \sigma(S_0, \ldots, S_{t-1})$ the filtration generated by past selection decisions. We focus on a restricted but reasonably rich class of *decision driven* rules:

**Definition 2.1.** A selection rule $\mathcal{S}_t : \mathcal{X} \to \{0, 1\}$ is called *decision driven* if $\mathcal{S}_t$ is $\mathcal{F}_{t-1}$-measurable.

For example, when $\mathcal{X} = \mathbb{R}$, the following selection rule is clearly decision driven, as it depends only on past selections $\{S_i\}_{i \leq t-1}$ and constants $\tau_0, \tau_1$:

$$x \mapsto \mathbb{1}\left\{x < \tau_1 + \frac{1}{\tau_0} \sum_{i=0}^{t-1} S_i\right\}.$$

The resulting sequence of selection decisions $\{S_t\}_{t \geq 0}$ is then defined recursively. At time $t = 0$, the threshold equals $\tau_1$, so $S_0 = 1$ if $X_0 < \tau_1$, and $S_0 = 0$ otherwise. At time $t = 1$, the threshold becomes $\tau_1 + S_0$; we then set $S_1 = 1$ if $X_1 < \tau_1 + S_0$, and $S_1 = 0$ otherwise. This recursive procedure continues analogously for all $t \geq 0$. For now, let $\widehat{C}_t : \mathcal{X} \times [0, 1] \to 2^{\mathcal{Y}}$ denote a generic prediction interval for $Y_t$, with details on its construction and validity addressed in the next section (for simplicity we omit its second argument corresponding to the target miscoverage level). Define $\{\widehat{C}_t(X_t) : S_t = 1\}_{t \geq 0}$ as the collection of prediction intervals produced by an online selective procedure (namely, only reporting $\widehat{C}_t(X_t)$ whenever $S_t = 1$).

Let $R : \mathcal{X} \times \mathcal{Y} \to \mathbb{R}$ denote a generic non-conformity score, such as absolute residuals $R_i = |\widehat{\mu}(X_i) - Y_i|$. We refer to $\{(X_i, Y_i) : i \in \mathcal{D}_t\}$ as calibration data at time $t$. For $\alpha \in (0, 1)$, the empirical quantile of the non-conformity scores $\{R_i\}_{i \in \mathcal{D}_t}$ is denoted by $\widehat{Q}_\alpha(\{R_i\}_{i \in \mathcal{D}_t})$.

Now, define the index set

$$\mathcal{J}_t = \mathcal{J}_{t,\text{off}} \cup \mathcal{J}_{t,\text{on}},$$

where $\mathcal{J}_{t,\text{off}} = \{-n, \dots, -1\}$, with $n \in \mathbb{N}$, is the set of indices of some *offline* data, and $\mathcal{J}_{t,\text{on}} = \{0, \dots, t-1\}$ the indices of the *online* data up to time $t$, respectively. Additionally, let $N = N_{\text{off}} + N_{\text{on}}$, where $N_{\text{off}} = |\mathcal{J}_{t,\text{off}}|$ and $N_{\text{on}} = |\mathcal{J}_{t,\text{on}}|$. Furthermore, we assume that the offline and online data are exchangeable, though this assumption will be relaxed later. Additionally, following Bao et al. (2024a), we make the following assumption through the paper:

(A) *All decision driven selection rules $\{\mathcal{S}_t\}_{t\geq 0}$ are independent of the offline data $\{Z_i\}_{i \in \mathcal{J}_{t,\text{off}}}$.*

We adopt Assumption (A)—that the selection rules are independent of the offline calibration pool—to mirror the setting of Bao et al. (2024a), who operate under the same assumption. This alignment enables a direct and transparent comparison between existing and novel calibration selection strategies. Importantly, if Assumption (A) is violated, post-selection exchangeability may no longer hold, and the formal validity guarantees presented in this paper do not apply in general. Finally, we note that Assumption (A) is necessary whenever offline data is used in the calibration selection process: without it, the selection rules may depend on the offline pool, breaking the permutation invariance required for our exchangeability arguments.

From time $t = 0$ onward data arrives sequentially, and selections are performed for this newly arriving data. Lastly, we formally introduce the online *calibration* (data) selection protocol. Let $\mathcal{K}_t : \mathcal{X} \to \{0, 1\}$ denote the *calibration selection strategy* (*strategy* for short). Consequently, we define the calibration selection *protocol* (mapping data to calibration indices) as

$$\mathcal{I}_t : \mathcal{Z}^{N+1} \to 2^{\mathcal{J}_t} : \quad (Z_{-n}, \dots, Z_t) \mapsto \{j \in \mathcal{J}_t : \mathcal{K}_t(X_j) = 1\}. \tag{P}$$

Thus, we have $\mathcal{D}_t \coloneqq \mathcal{I}_t(Z_{-n}, \dots, Z_t) \subseteq \mathcal{J}_t$. Specifically, $\mathcal{K}_t(X_j) = 1$ indicates that, at time $t$, the candidate calibration datum $Z_j = (X_j, Y_j)$ will be included in the calibration set. Note that, we only perform calibration selection, if $\mathcal{S}_t(X_t) = 1$. Otherwise, calibration data is unnecessary, as prediction intervals are constructed—and thus require a calibration set—only when the test datum is selected. Importantly, different choices of $\mathcal{K}_t$ influence both the size and composition of the calibration set, which in turn impacts the validity and informativeness of the resulting prediction intervals.

## 3 Online Selective Conformal Inference

An online selective (split) conformal inference procedure involves the following steps. Up to time $t \geq 0$, we observe the data $(X_0, Y_0), \dots, (X_{t-1}, Y_{t-1})$, which consists of feature-response pairs from previous time points. Then, in the vein of split conformal prediction, we proceed as follows (Bao et al., 2024a):

1. Specify the selection rule $\mathcal{S}_t \in \mathcal{F}_{t-1}$, and calibration selection strategy $\mathcal{K}_t$ *before* observing the current feature $X_t$.

2. Observe $X_t$, and if $S_t = 1$ decide to report a prediction interval for the yet unobserved $Y_t$.

3. Obtain the calibration indices $\mathcal{D}_t$ via (P).

4. Compute the non-conformity scores $\{R_i\}_{i \in \mathcal{D}_t}$ for the calibration set.

5. Report the prediction interval

$$\widehat{C}_t(X_t) = \{y \in \mathcal{Y} : R(X_t, y) \leq \widehat{Q}_{1-\alpha}(\{R_i\}_{i \in \mathcal{D}_t})\}.$$

A crucial aspect to address is the *selection* of calibration data. In standard (online) conformal prediction, the calibration data is typically chosen as $\mathcal{D}_t = \mathcal{J}_t$, which corresponds to choosing the calibration selection strategy $\mathcal{K}_t(X_i) \equiv 1$ in the calibration selection protocol (P). However, using the *entire* available data for calibration poses challenges in the selective setting. Roughly speaking, the selection rule introduces an asymmetric dependence between the current data point and past data through the selection process.

Recall that we focus on decision driven selection rules; while the selection *rule* itself does not depend on $X_t$, it does rely on past *decisions* and therefore depends on $\{X_i\}_{i \in \mathcal{J}_t}$. Such dependencies typically violate exchangeability, the workhorse of conformal prediction. As a result, naively applying standard conformal methods can lead to violations of desired coverage guarantees. Mitigating this requires carefully designing calibration selection strategies that account for the dependence introduced by the selection mechanism. The challenge, therefore, is to "design" calibration selection strategies that preserve exchangeability while maintaining a sufficiently large calibration set. As we will see soon, different strategies impose varying trade-offs between statistical guarantees and practical applicability. Some approaches prioritize preserving exchangeability at the cost of reducing the calibration set size, while others aim to retain more calibration points but risk violating exchangeability.

A natural question that arises is: given that a test point has been selected, can we identify a suitable calibration selection strategy that *ensures* exchangeability between the calibration data and the test datum? The answer is affirmative but also pinpoints the challenges inherent in the selective setting.

### 3.1 Existing Calibration Selection Strategies

To address this question, we first discuss both existing and novel calibration selection strategies. A natural point of departure is the *full* strategy (`FULL`):

$$\mathcal{K}_t(X_j) = 1, \text{ for } j \in \mathcal{J}_t . \tag{$i$}$$

In other words, at each selected time the calibration data consists of *all* previously observed data, i.e., $\mathcal{D}_t = \mathcal{J}_t$. While this strategy yields the largest calibration set, it typically breaks exchangeability because the test datum is selected via a rule, whereas the calibration set is unaffected by the rule, consisting of all previous observations.

Another strategy, the *selection-full* strategy (`S-FULL`) selects calibration data according to the selection rule given at time $t$, i.e.,

$$\mathcal{K}_t(X_j) = \mathcal{S}_t(X_j), \text{ for } j \in \mathcal{J}_t . \tag{$ii$}$$

However, because we restrict ourselves to decision driven selection rules, the test datum at time $t$ is selected independently of future outcomes, while this is not true of the calibration set. This asymmetric dependency leads to a violation of exchangeability.

Similarly, we define the *selection-fixed* strategy (`S-FIX`), which selects calibration data (according to the selection rule given at time $t$) solely from the offline data:

$$\mathcal{K}_t(X_j) = \begin{cases} \mathcal{S}_t(X_j) & \text{for } j \in \mathcal{J}_{t,\text{off}}, \\ 0 & \text{for } j \in \mathcal{J}_{t,\text{on}} . \end{cases} \tag{$iii$}$$

If Assumption (A) holds, this strategy is unproblematic, as the offline data remains independent of the selection rules. However, caveats exist: when no offline calibration data is available or when Assumption (A) does not hold. Additionally, even if Assumption (A) is satisfied, very few calibration points may be selected, no matter how large $t$ is.

Recognizing these limitations, Bao et al. (2024a) recently introduced an alternative strategy. The *adaptive* calibration selection strategy (`ADA`) is defined as follows:

$$\mathcal{K}_t(X_j) = \begin{cases} \mathcal{S}_t(X_j) & \text{for } j \in \mathcal{J}_{t,\text{off}}, \\ \mathcal{S}_t(X_j)\mathbb{1}\{\mathcal{S}_j(X_j) = \mathcal{S}_j(X_t)\} & \text{for } j \in \mathcal{J}_{t,\text{on}} . \end{cases} \tag{$iv$}$$

Bao et al. (2024a) claim that this adaptive calibration selection strategy ensures selection-conditional coverage. However, we soon provide both theoretical and empirical evidence demonstrating that this is, in fact, *not* the case.

### 3.2 Novel Calibration Selection Strategies

To address such issues, we introduce the *exchangeability-preserving* strategy (`EXPRESS`), defined as follows:

$$\mathcal{K}_t(X_j) = \mathcal{S}_t(X_j) \prod_{i \in \mathcal{J}_{t,\text{on}}} \mathbb{1}\{\mathcal{S}_i(X_j) = \mathcal{S}_i(X_t)\}, \text{ for } j \in \mathcal{J}_t. \quad (v)$$

Clearly, this strategy is quite stringent, as it requires identical past selection decisions for both the candidate calibration point and the selected test datum. Consequently, the calibration data includes only points that have undergone the same selection process as the test datum, strictly preserving exchangeability. However, this restrictiveness can significantly reduce the calibration set size, potentially rendering it empty in some cases.

To relax the strict constraints of the previous strategy, we introduce a variant called the *k-exchangeable preserving* strategy (`K-EXPRESS`), which considers only the last $k \geq 1$ points:

$$\mathcal{K}_t(X_j) = \mathcal{S}_t(X_j) \prod_{i=t-k}^{t-1} \mathbb{1}\{\mathcal{S}_i(X_j) = \mathcal{S}_i(X_t)\}, \text{ for } j \in \mathcal{J}_t^k, \quad (vi)$$

where $\mathcal{J}_t^k = \mathcal{J}_{t,\text{off}} \cup \{t-k, \ldots, t-1\}$. Instead of enforcing consistency with *all* past selection decisions, `K-EXPRESS` restricts the comparison to the last $k$ points. This variant mitigates the risk of an excessively small calibration set while still preserving exchangeability.

Our final strategy, called `EXPRESS-M`, applies both `S-FIX` and `EXPRESS` separately and then *merges* the resulting prediction intervals. That is, prediction intervals are constructed independently using both strategies, and their intersection is taken. While various merging schemes are possible, we use an uneven split of $\alpha$ between the two methods for constructing the final prediction interval. Specifically, for $t > 0$ we allocate $(1/\sqrt{t})\alpha$ to `S-FIX`, while the remaining $(1 - 1/\sqrt{t})\alpha$ is assigned to `EXPRESS`. To make this construction explicit, let $\alpha_{\text{SF}} = (1/\sqrt{t})\,\alpha$ and $\alpha_{\text{EX}} = (1 - 1/\sqrt{t})\,\alpha$, so that $\alpha_{\text{SF}} + \alpha_{\text{EX}} = \alpha$. At each time $t$, we construct intervals $\widehat{C}_t^{\text{SF}}$ and $\widehat{C}_t^{\text{EX}}$ at levels $1 - \alpha_{\text{SF}}$ and $1 - \alpha_{\text{EX}}$, respectively, and define the final prediction interval as

$$\widehat{C}_t^{\text{M}} = \widehat{C}_t^{\text{SF}} \cap \widehat{C}_t^{\text{EX}}.$$

This merging approach mitigates the restrictiveness of `EXPRESS`—which can result in too few calibration data and, consequently, excessively large prediction intervals—by leveraging the more inclusive nature of `S-FIX`.

*Remark* 3.1. One might naturally ask why we do not adopt a strategy that applies $\mathcal{S}_t(X_j)$ for $j \in \mathcal{J}_{t,\text{off}}$ while applying the exchangeability-preserving strategy $(v)$ for $j \in \mathcal{J}_{t,\text{on}}$. This strategy would indeed be less restrictive, and therefore yield a larger calibration set. While this may seem unproblematic at first glance, we show that it subtly violates the exchangeability between the (offline) calibration data and the test datum.

## 4 Selection-Conditional Coverage

In this section, we show shortcomings of existing calibration selection strategies in ensuring selection-conditional coverage. Additionally, we demonstrate that online selective conformal inference, when instantiated with our novel strategies, guarantees selection-conditional coverage. First, we evaluate both existing and novel calibration selection strategies empirically with the following simulation:

**Simulation 4.1.** In each iteration of the simulation, we generate data of size $N = N_{\text{off}} + N_{\text{on}}$. For the following results, we set $N_{\text{off}} = 10$ and $N_{\text{on}} = 20$. We generate a univariate feature $X \sim \text{Unif}[0,2]$ and model the response as

$$Y = \mu(X) + \epsilon \quad \text{with} \quad \mu(X) = X\beta,$$

where we assume a heterogeneous noise distribution, i.e., $\epsilon \mid X \sim \mathcal{N}(0, X/2)$. For simplicity, we assume $\beta = 1$. Since we are in a synthetic setting, we have direct access to the true function and use it as our model;

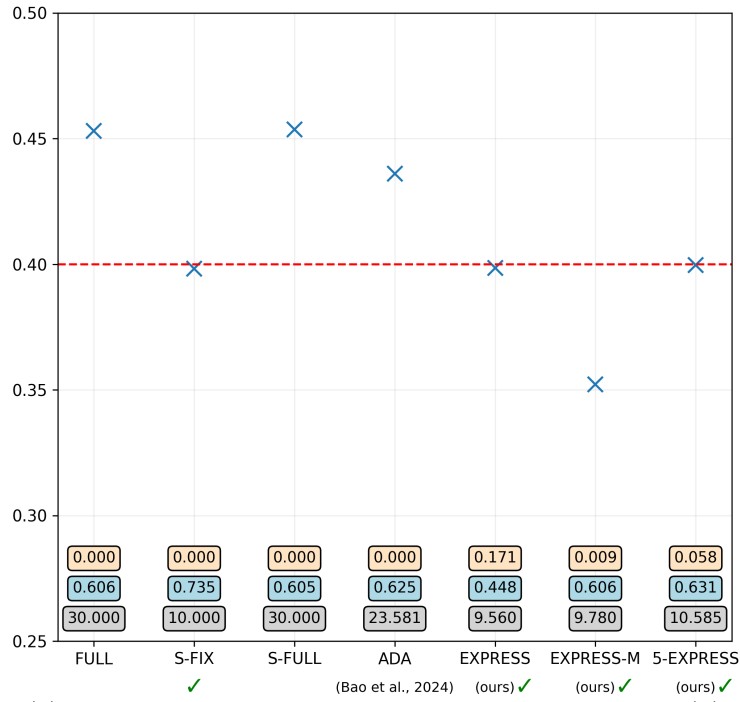

Figure 2: Miscoverage ($\times$) is shown alongside the number of calibration points (■), median interval length (■) and the fraction of infinite length prediction intervals (■). We highlight provably correct methods (✔) and the target level (- -). *All metrics are averaged over $N = 1 \times 10^6$ runs.*

specifically, we have $\widehat{\mu}(\cdot) = \mu(\cdot)$. Further, the selection rules are defined as

$$x \mapsto \begin{cases} \mathbb{1}\left\{x < 1 + \frac{1}{\tau_0}\sum_{i=0}^{j-1} S_i\right\} & \text{for } j \leq t-1 \\ \mathbb{1}\left\{\sum_{i=0}^{j-1} S_i > \tau_1\right\} & \text{for } j = t, \end{cases}$$

where we choose $\tau_0 = 20$ and $\tau_1 = 16$. We then perform online selective conformal prediction with both existing and novel strategies. All reported metrics are averaged over $N = 1 \times 10^6$ runs.

*Results.* Figure 2 summarizes the performance of different calibration selection strategies in online selective conformal prediction. With the exception of `S-FIX`, all existing strategies *fail* to provide valid selection-conditional prediction intervals, as they do not maintain the target miscoverage level of $\alpha = 0.4$. In contrast, both `S-FIX` and the family of exchangeability-preserving strategies achieve exact coverage. From a practical standpoint, while `EXPRESS` yields the smallest prediction intervals, its restrictive nature results in a significant fraction ($\approx 17.71\%$) of infinite-length intervals. Its variants, `EXPRESS-M` and `K-EXPRESS`, mitigate this issue and perform comparably well.

These results empirically highlight the fundamental shortcomings of existing calibration selection strategies in ensuring selection-conditional validity. To reiterate, the exchangeability-preserving approaches not only achieve the desired theoretical guarantees but also offer practical advantages by balancing interval tightness and calibration set size. While `EXPRESS` is the most conservative, its variants effectively address its limitations, making them viable alternatives in real-world applications. In Appendix D, we present additional simulations using alternative decision driven selection rules.

We now proceed to the theoretical results of this section. First, we define the *selection* procedure, which maps data to the augmented calibration indices

$$\widetilde{\mathcal{I}}_t : \mathcal{Z}^{N+1} \to 2^{\mathcal{J}_t \cup \{t\}} : (Z_{-n}, \ldots, Z_t) \mapsto \begin{cases} \mathcal{I}_t(Z_{-n}, \ldots, Z_t) \cup \{t\} & \text{if } \mathcal{S}_t(X_t) = 1, \\ \mathcal{I}_t(Z_{-n}, \ldots, Z_t) & \text{otherwise.} \end{cases}$$

where $\mathcal{I}_t(\cdot)$ denotes the calibration selection protocol (P). So

$$t \in \widetilde{\mathcal{I}}_t \text{ if and only if } \mathcal{S}_t(X_t) = 1. \tag{4}$$

Now, let $\widetilde{\mathcal{D}}_t := \widetilde{\mathcal{I}}_t(Z_{-n}, \ldots, Z_t)$ and let $\pi$ be a permutation of the indices in $\widetilde{\mathcal{D}}_t$. Define the extended permutation of the indices in $\{-n, \ldots, t\}$ as

$$\widetilde{\pi}(i) = \begin{cases} \pi(i) & \text{for } i \in \widetilde{\mathcal{D}}_t, \\ i & \text{for } i \notin \widetilde{\mathcal{D}}_t. \end{cases} \tag{5}$$

Whether or not $\{Z_i\}_{i \in \widetilde{\mathcal{D}}_t}$ (i.e., the selected calibration data, and the selected test datum) are exchangeable depends on the calibration selection protocol (P). Recall that $\{Z_i\}_{i \in \widetilde{\mathcal{D}}_t}$ are exchangeable if for any set of indices $D$ such that $\mathbb{P}(\widetilde{\mathcal{D}}_t = D) > 0$, and any permutation $\pi$ of indices in $D$ (equivalently, any extended permutation $\widetilde{\pi}$ as in (5)), and any measurable $A \subseteq (\mathcal{X} \times \mathcal{Y})^{|D|}$, we have

$$\mathbb{P}\{(Z_i)_{i \in D} \in A \,|\, \widetilde{\mathcal{D}}_t = D\} = \mathbb{P}\{(Z_{\widetilde{\pi}(i)})_{i \in D} \in A \,|\, \widetilde{\mathcal{D}}_t = D\}. \tag{6}$$

To demonstrate (6), it is sufficient to show that

$$\mathbb{P}\{(Z_i)_{i \in D} \in A \,, \widetilde{\mathcal{I}}_t(Z_{-n}, \ldots, Z_t) = D\} = \mathbb{P}\{(Z_{\widetilde{\pi}(i)})_{i \in D} \in A \,, \widetilde{\mathcal{I}}_t(Z_{\widetilde{\pi}(-n)}, \ldots, Z_{\widetilde{\pi}(t)}) = D\}$$

$$\overset{(\star)}{=} \mathbb{P}\{(Z_{\widetilde{\pi}(i)})_{i \in D} \in A \,, \widetilde{\mathcal{I}}_t(Z_{-n}, \ldots, Z_t) = D\}.$$

While the first equality follows from exchangeability of $(Z_{-n}, \ldots, Z_t)$, the important argument lies in step $(\star)$. Specifically, $(\star)$ holds true if

$$\widetilde{\mathcal{I}}_t(Z_{-n}, \ldots, Z_t) = \widetilde{\mathcal{I}}_t(Z_{\widetilde{\pi}(-n)}, \ldots, Z_{\widetilde{\pi}(t)}). \tag{S}$$

**Lemma 4.2.** *Let* $\{\mathcal{S}_t\}_{t \geq 0}$ *be a sequence of decision driven selection rules and denote by* $\{\mathcal{S}_t^{\widetilde{\pi}}\}_{t \geq 0}$ *the sequence of selection rules generated when operating on* $Z_{\widetilde{\pi}(-n)}, \ldots, Z_{\widetilde{\pi}(t)}$. *The selection rule sequences* $\{\mathcal{S}_t\}_{t \geq 0}$ *and* $\{\mathcal{S}_t^{\widetilde{\pi}}\}_{t \geq 0}$ *are identical, if* $\mathcal{S}_j(X_j) = \mathcal{S}_j(X_{\widetilde{\pi}(j)})$ *for all* $j \in \mathcal{J}_{t,\text{on}}$.

*Proof.* Assume $\mathcal{S}_j(X_j) = \mathcal{S}_j(X_{\widetilde{\pi}(j)})$ for all $j \in \mathcal{J}_{t,\text{on}}$. Now, let $j = 0$. By assumption, we have $\mathcal{S}_0(X_0) = \mathcal{S}_0(X_{\widetilde{\pi}(0)})$. This yields $\mathcal{S}_1^{\widetilde{\pi}} \equiv \mathcal{S}_1$; in other words the *rules* at time $t = 1$ are identical. Then, the claim follows by induction. This completes the proof. $\qquad \square$

The following lemma demonstrates that the selection procedure, when instantiated with strategies $(i)$, $(ii)$, or $(iv)$, does not necessarily satisfy symmetry (S). While this may be apparent for the full and selection-full calibration selection strategy, we explicitly demonstrate that symmetry also *fails* for the recently proposed adaptive rule.

**Lemma 4.3.** *Without additional assumptions, the selection strategies (*i*), (*ii*), and (*iv*) do not satisfy symmetry (S).*

*Proof.* Let $\{\mathcal{S}_t^{\widetilde{\pi}}\}_{t \geq 0}$ be the sequence of selection rules generated when operating on $Z_{\widetilde{\pi}(-n)}, \ldots, Z_{\widetilde{\pi}(t)}$. Now, let the permutation $\pi$ be a transposition such that $\pi(t) = s$, $\pi(s) = t$ and $\pi(j) = j$ for $j \neq s, t$. Fix $j \in \widetilde{\mathcal{D}}_t \cap \mathcal{J}_{t,\text{on}}$. For strategies $(i)$, and $(ii)$ the claim follows immediately, since we have $\mathcal{S}_t(X_t) \neq \mathcal{S}_t^{\widetilde{\pi}}(X_s)$ without additional assumptions. By definition of strategy $(iv)$, $\mathcal{S}_t(X_j)\mathbb{1}\{\mathcal{S}_j(X_j) = \mathcal{S}_j(X_t)\} = 1$. If symmetry (S) holds, we expect that

$$\mathcal{S}_t^{\widetilde{\pi}}(X_{\pi(j)})\mathbb{1}\{\mathcal{S}_j^{\widetilde{\pi}}(X_{\pi(j)}) = \mathcal{S}_j^{\widetilde{\pi}}(X_{\pi(t)})\} = 1. \tag{7}$$

Again, by definition of $(iv)$, we have $\mathcal{S}_s(X_s) = \mathcal{S}_s(X_t)$. Hence, Lemma 4.2 ensures that the selection sequences $\{\mathcal{S}_t\}_{t\geq 0}$ and $\{\mathcal{S}_t^{\widetilde{\pi}}\}_{t\geq 0}$ are indeed identical. Consequently, Equation (7) can be reformulated as

$$\mathcal{S}_t(X_{\pi(j)})\mathbb{1}\{\mathcal{S}_j(X_{\pi(j)}) = \mathcal{S}_j(X_{\pi(t)})\} = 1. \tag{8}$$

Given that $\pi(j) \in \widetilde{\mathcal{D}}_t$, it follows by definition of the extended permutation that $\mathcal{S}_t(X_{\pi(j)}) = 1$. However, the issue in (8) lies in the fact that

$$\mathcal{S}_j(X_{\pi(j)}) = \mathcal{S}_j(X_{\pi(t)})$$

*cannot* be guaranteed without additional assumptions. This completes the proof. $\qquad\square$

What Lemma 4.3 shows is that the selection strategies $(i)$, $(ii)$, and $(iv)$ do not satisfy symmetry (S), in general. However, note that (S) is a *sufficient* but not *necessary* condition for $(\star)$ in the above display. This leaves open the possibility that these strategies might still satisfy $(\star)$ in certain cases. Nevertheless, as demonstrated by the counterexample in Simulation 4.1, they do not satisfy it in general.

Now we turn to our positive results. Specifically, we show that the exchangeability-preserving strategies satisfy symmetry, thereby implying exchangeability and, via classical conformal arguments (as demonstrated in Appendix B), selection-conditional coverage.

**Lemma 4.4.** *The selection procedure instantiated with strategies (*iii*), (*v*) or (*vi*) satisfies symmetry (S). Thus, the random variables $\{Z_i\}_{i\in\widetilde{\mathcal{D}}_t}$ are exchangeable.*

*Proof.* Due to Assumption (A) the claim for strategy $(iii)$ holds trivially true. For strategy $(v)$, consider any arbitrary permutation $\pi$ and let $j \in \widetilde{\mathcal{D}}_t$. By definition, it follows that

$$\mathcal{S}_t(X_j)\prod_{i=0}^{t-1}\mathbb{1}\{\mathcal{S}_i(X_j) = \mathcal{S}_i(X_t)\} = 1, \tag{9}$$

meaning that all the above indicators are equal to one. Applying the same observation to $k \in \widetilde{\mathcal{D}}_t$, we see that

$$\forall j, k \in \widetilde{\mathcal{D}}_t : \ \forall i \geq 0 : \mathcal{S}_i(X_j) = \mathcal{S}_i(X_k) = \mathcal{S}_i(X_t). \tag{10}$$

Then, by Lemma 4.2 we know that the selection rule sequences $\{\mathcal{S}_t\}_{t\geq 0}$ and $\{\mathcal{S}_t^{\widetilde{\pi}}\}_{t\geq 0}$ are identical. Consequently, we can rewrite (9) as follows:

$$\mathcal{S}_t^{\widetilde{\pi}}(X_j)\prod_{i=0}^{t-1}\mathbb{1}\{\mathcal{S}_i^{\widetilde{\pi}}(X_j) = \mathcal{S}_i^{\widetilde{\pi}}(X_t)\} = 1.$$

Since $\pi(j) \in \widetilde{\mathcal{D}}_t$, we also have $S_i(X_j) = S_i(X_{\pi(j)})$ from (10). Thus,

$$\mathcal{S}_t^{\widetilde{\pi}}(X_{\pi(j)})\prod_{i=0}^{t-1}\mathbb{1}\{\mathcal{S}_i^{\widetilde{\pi}}(X_{\pi(j)}) = \mathcal{S}_i^{\widetilde{\pi}}(X_{\pi(t)})\} = 1.$$

Analogous reasoning yields the claim for strategy $(vi)$. Since (S) is sufficient for $(\star)$, the random variables $\{Z_i\}_{i\in\widetilde{\mathcal{D}}_t}$ are exchangeable. This completes the proof. $\qquad\square$

In fact, we can show that online conformal inference with the exchangeability-preserving strategies ensures an even *stronger* notion of selection-conditional coverage:

**Theorem 4.5.** *Online selective conformal inference with strategies (*iii*), (*v*) or (*vi*) produces strong selection-conditional prediction intervals. Specifically, we have*

$$\mathbb{P}\left\{Y_t \in \widehat{C}_t(X_t) \mid S_0 = s_0, \ldots, S_{t-1} = s_{t-1}, S_t = 1\right\} \geq 1 - \alpha$$

*for any $t \geq 0$ and any $(s_0, \ldots, s_{t-1}) \in \{0,1\}^{t-1}$, when $\mathbb{P}(S_0 = s_0, \ldots, S_{t-1} = s_{t-1}, S_t = 1) > 0$. Thus, selection-conditional coverage (2) is fulfilled.*

Theorem 4.5 is proved in Appendix B. Since Theorem 4.5 ensures (strong) selection-conditional validity for strategies (*iii*) and (*v*), applying the Bonferroni inequality (Bonferroni, 1936) guarantees that the intersection of the corresponding prediction intervals remains valid. Hence, the *merging* strategy also produces (strong) selection-conditional prediction intervals.

To summarize, we have shown that while (most) existing strategies fail to ensure selection-conditional coverage, our novel proposals successfully achieve it. An equally important takeaway is that achieving selection-conditional coverage is *nontrivial*, as exchangeability can be violated in many ways, necessitating a fully symmetric strategy – one that, as we have shown, can be quite restrictive (in terms of selecting suitable calibration data).

## 5 FCR Control

Another metric for evaluating the errors in reported prediction intervals is the false coverage rate (FCR), as defined in (3). We evaluate whether online selective conformal prediction, with both existing and novel strategies, effectively controls the FCR below a specified target level.

**Simulation 5.1.** The data generation follows the same setup as in Simulation 4.1. Here, however, we consider a longer time horizon by choosing $N_{\mathrm{off}} = 50$ and $N_{\mathrm{on}} = 200$. For $t \geq 0$, the selection rule is given by

$$x \mapsto \mathbb{1}\left\{x < \tau_1 + \frac{1}{\tau_0}\sum_{i=0}^{t-1} S_i\right\}, \tag{11}$$

where we choose $\tau_0 = 200$ and $\tau_1 = 1$.

*Results.* Figure 3 provides a comparison of existing and novel strategies in terms of FCR control, growth of calibration sets, and the informativeness of prediction intervals over time. We discuss the results in the following:

Subplot **(a)** illustrates the FCR as time progresses. All novel strategies successfully control the FCR at the target level $\alpha = 0.4$, while `ADA` fails to do so. Although other existing strategies appear to control FCR in this specific simulation, in Appendix D we present cases where they do not. This highlights a key insight: depending on the data-generating process and selection rule, some strategies may incidentally control FCR, even though they do not provably do so.

Subplot **(b)** reports the number of calibration points over time. As expected, all strategies exhibit their characteristic behavior: `S-FIX` selects calibration points exclusively from the offline set, preventing any growth beyond its initial size $N_{\mathrm{off}} = 50$, while `FULL` and `S-FULL` accumulate calibration points unrestrictedly. Notably, `EXPRESS-M` emerges as a compromise between `EXPRESS` and `S-FIX`, balancing the benefits of a more restrictive approach that leverages online data with the need for a sufficiently large calibration set to construct informative prediction intervals.

Subplot **(c)** reports the fraction of prediction intervals that are of infinite length. The `EXPRESS` strategy exhibits a steadily increasing fraction of such intervals over time, highlighting its restrictive nature, which often leaves too few calibration points to construct finite prediction intervals. In contrast, variants such as `EXPRESS-M` and `10-EXPRESS`, demonstrate substantial improvements. This highlights their viability as more flexible alternatives that control FCR while ensuring the prediction intervals remain informative.

Subplot **(d)** shows the median length of prediction intervals as a function of time. Although `FULL` accumulates the largest calibration set, it produces the widest intervals because it includes all previous examples without considering the selection mechanism, leading clearly to selection bias. In this simulation, `S-FULL`, `S-FIX`, `10-EXPRESS`, and `ADA` all exhibit a steady increase in median interval size. By contrast, `EXPRESS-M` gradually converges to the behavior of `EXPRESS`, which maintains the shortest intervals overall. However, it is important to note that `EXPRESS` incurs an increasing fraction of infinite intervals (as seen in subplot **(c)**), which is not reflected by the median lengths. In contrast, `EXPRESS-M` manages to avoid these infinite intervals while still preserving relatively short intervals, providing a more practical alternative.

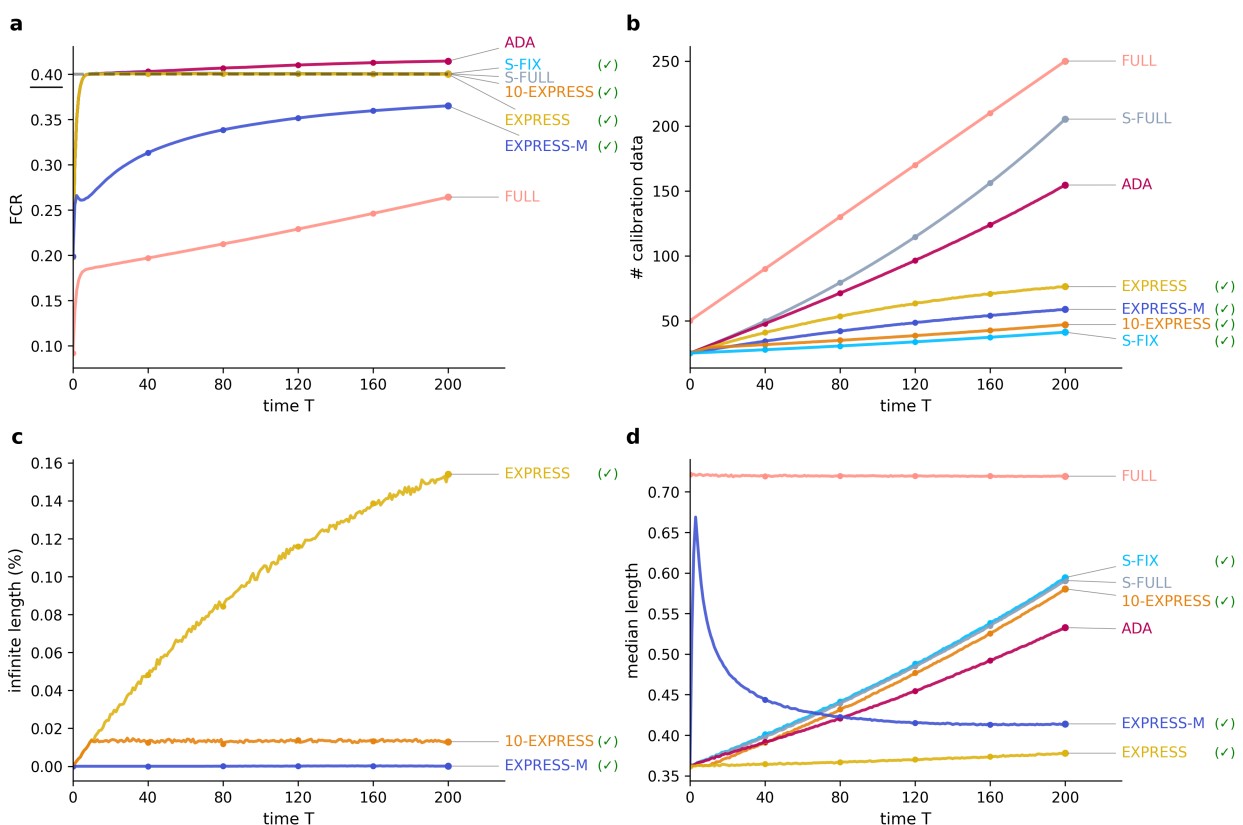

Figure 3: **(a)** FCR as a function of time $T$. The dashed black line represents the target level $\alpha = 0.4$. We highlight provably correct methods (✔). **(b)** Number of calibration data points used over time. Strategies accumulating more calibration data tend to yield shorter prediction intervals. **(c)** Fraction of prediction intervals of infinite length over time. A high fraction suggests a strategy often fails to provide informative intervals. Only reported for novel strategies. **(d)** Median prediction interval length over time. Shorter intervals indicate higher informativeness. *All metrics are averaged over $N = 1 \times 10^4$ runs.*

Overall, these results highlight the trade-offs in calibration selection strategies. Methods that strictly enforce exchangeability, such as `EXPRESS`, offer theoretical guarantees but at the cost of increasingly frequent infinite-length intervals. More flexible approaches like `EXPRESS-M` and `10-EXPRESS` mitigate this issue by maintaining a larger calibration set while still controlling FCR. Notably, strategies that ignore selection bias, such as `FULL`, `S-FULL`, and `ADA` may appear reasonable in some cases but lack theoretical guarantees, making them unreliable in general.

In fact, even the conditional expectation of FCP given that at least one selection is made,

$$\mathrm{pFCR}(T) = \mathbb{E}\left[\mathrm{FCP}(T) \,\middle|\, \sum_i S_i > 0\right] \tag{12}$$

is controlled when using our proposed strategies. The above metric is called *positive* FCR (Weinstein & Ramdas, 2020), in analogy to the positive false discovery rate (FDR) (Storey, 2003).

**Proposition 5.2.** *Strong selection-conditional coverage implies that* $\mathrm{pFCR}(T) \leq \alpha$ *for any* $T \geq 0$. *Moreover, if strong selection-conditional coverage holds exactly, then* $\mathrm{pFCR}(T) = \alpha$. *Thus, calibration selection strategies (*iii*), (*v*) or (*vi*) guarantee that* $\mathrm{FCR}(T) \leq \alpha$ *for any* $T \geq 0$. *If* $\sum_i S_i > 0$ *holds almost surely, then* $\mathrm{FCR}(T) = \alpha$ *under the same exact coverage condition.*

With similar arguments as for (strong) selection-conditional coverage, it follows that the *merging* strategy also controls the FCR at a user-defined target level.

# 6 Other Conformal Methods

Our discussion so far was only concerned about online selective conformal prediction algorithm in the sense of Section 3. In principle, other conformal methods can also be applied to this problem. We first outline two of those: conformal LORD-CI (Weinstein & Ramdas, 2020) and adaptive conformal inference (ACI) (Gibbs & Candes, 2021), and discuss (potential) use-cases of such algorithms. We give more details about these methods in Appendix C.

We also note that a related discussion can be found in Bao et al. (2024a). However, we want to highlight a crucial distinction: while Bao et al. (2024a) consider adaptive conformal inference algorithms to address scenarios where the exchangeability of the underlying data itself is violated, our perspective differs. We argue that when the selection mechanism disrupts exchangeability between the test datum and the calibration data, this too constitutes a form of distribution shift—induced not by the data-generating process itself, but by the selective nature of the inference procedure.

## 6.1 Conformal LORD-CI

The LORD-CI algorithm was originally proposed by Weinstein & Ramdas (2020). Let $\alpha_t \in (0,1)$ be a $\mathcal{F}_{t-1}$-measurable coverage level, then a conformal prediction interval is constructed as

$$\widehat{C}_t(X_t) = \{y \in \mathcal{Y} : R(X_t, y) \leq \widehat{Q}_{1-\alpha_t}(\{R_i\}_{i \in \mathcal{J}_{t,\mathrm{off}}})\}. \tag{13}$$

The marginal level $\alpha_t$ is updated dynamically by maintaining the *invariant*

$$\frac{\sum_{t=0}^{T} \alpha_t}{1 \vee \sum_{t=0}^{T} S_t} \leq \alpha, \quad \text{for any } T \geq 0. \tag{14}$$

Any online protocol maintaining the invariant (14) is called (conformal) LORD-CI procedure.

**Proposition 6.1.** *Any conformal LORD-CI procedure with calibration indices $\mathcal{D}_t = \mathcal{J}_{t,\mathrm{off}}$ ensures that* FCR$(T) \leq \alpha$ *for all* $T \geq 0$.

The constraint $\mathcal{D}_t = \mathcal{J}_{t,\mathrm{off}}$ is critical, and causes the method to be conservative in practice. We are presently unsure if the result can be expanded to allow utilizing $\mathcal{J}_{t,\mathrm{on}}$.

## 6.2 Adaptive Conformal Inference

Adaptive conformal inference (ACI) (Gibbs & Candes, 2021) is a widely used conformal prediction algorithm, particularly suited for scenarios where exchangeability is violated, such as online settings with distribution shifts. The ACI algorithm adjusts the miscoverage level based on historical under- or over-coverage feedback. Specifically, for a target miscoverage level $\alpha$, it updates $\alpha_t$ according to

$$\alpha_t = \alpha_{t-1} + \gamma \left( \alpha - \mathbb{1}\{Y_{t-1} \notin \widehat{C}_t(X_{t-1}, \alpha_{t-1})\} \right),$$

where $\gamma > 0$ is a fixed step-size parameter. Here, $\widehat{C}_t(X_{t-1}, \alpha_{t-1})$ denotes the prediction interval constructed at time $t-1$ with nominal miscoverage level $\alpha_{t-1}$. This adaptive scheme *increases* $\alpha_t$ whenever the previous prediction interval fails to cover $Y_{t-1}$ (indicating under-coverage) and *decreases* $\alpha_t$ otherwise (indicating over-coverage), thereby steering the empirical miscoverage rate toward the specified target $\alpha$.

Recently, Gibbs & Candès (2024) proposed an extension of ACI, termed DtACI, by employing an exponential re-weighting scheme to estimate the parameter $\gamma$. Another recent work in this regard is Angelopoulos et al. (2024b).

**Proposition 6.2** (adapted, Gibbs & Candes (2021)). *For all $T \geq 0$ we have that*

$$|\,\mathrm{FCR}(T) - \alpha| \leq \frac{\max\{\alpha_1, 1 - \alpha_1\} + \gamma}{\sum_i S_i \gamma} \tag{15}$$

*almost surely. Particularly,* $\lim_{T \to \infty} \mathrm{FCR}(T) \overset{\mathrm{a.s.}}{=} \alpha$.

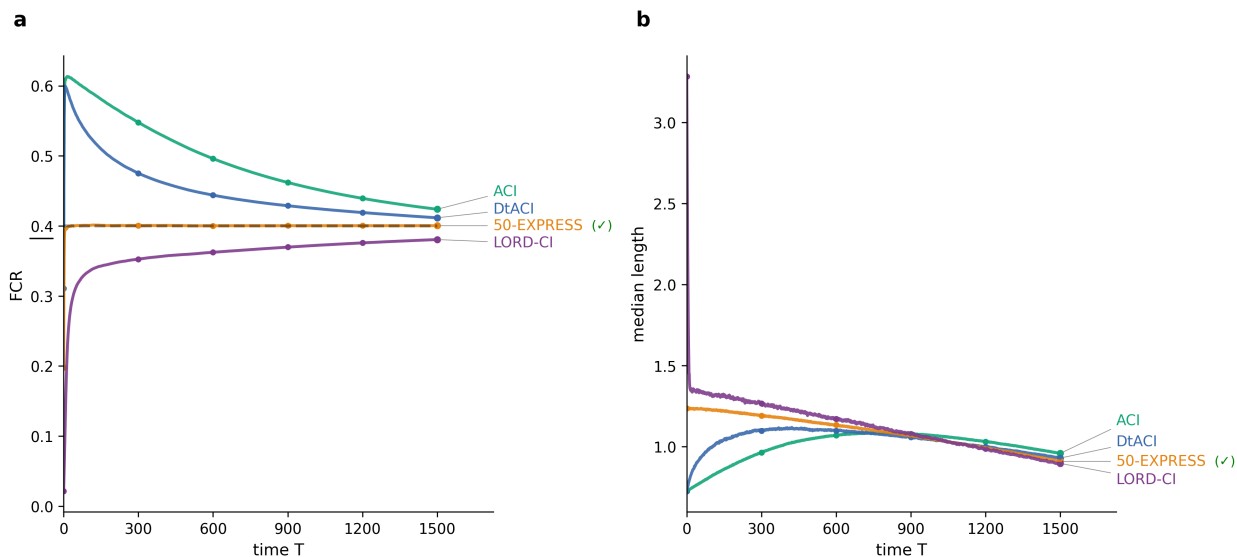

Figure 4: **(a)** FCR as a function of time $T$. The dashed black line represents the target level $\alpha = 0.4$. We highlight provably correct methods (✔). **(b)** Median prediction interval length over time. Shorter intervals indicate higher informativeness. *All metrics are averaged over $N = 1 \times 10^4$ runs.*

The proof of Proposition 6.2 follows directly from Gibbs & Candes (2021, Proposition 4.1). We also note that Bao et al. (2024a) established a similar result for DtACI adapted to the selective setting.

**Simulation 6.3.** The setup is the same as in Simulation 5.1, with the only difference that we choose here $N_{\text{off}} = 200$ and $N_{\text{on}} = 1500$. For $t \geq 0$, the selection rule is again given by (11) where we choose here $\tau_0 = 1500$ and $\tau_1 = 1$.

*Results.* Figure 4 compares other conformal methods with online selective conformal prediction using the `50-EXPRESS` strategy in terms of FCR control and median prediction interval length over time.

Subplot **(a)** illustrates the FCR as time progresses. ACI and DtACI gradually approach the nominal target as time increases—albeit only asymptotically—while LORD-CI strictly controls FCR for all $T \geq 0$, but only in a notably conservative manner.

Subplot **(b)** shows the median length of prediction intervals over time. Here, LORD-CI starts off with the largest intervals, mirroring its conservative nature in FCR control. Over time, however, all methods converge toward a similar interval size, suggesting that while they differ in short-term behavior, their long-term median interval lengths stabilize to comparable levels.

While LORD-CI offers finite-sample FCR control, it does so at the cost of conservativeness. In contrast, adaptive approaches such as ACI and DtACI produce shorter intervals but only guarantee FCR control asymptotically.

As already noted by Bao et al. (2024a), ACI is particularly useful when the data is not exchangeable. While this often holds in online settings, this comes at the cost of forgoing selection-conditional coverage guarantees and finite FCR control (since only asymptotic FCR control would be guaranteed) when the data is exchangeable. On the positive side, ACI can be applied directly without modification, making it a convenient off-the-shelf solution. However, this convenience comes at the expense of leveraging the exchangeability of the data (when this is the case), which could otherwise be exploited for stronger theoretical guarantees.

## 7 Other Related Work

As previously noted, our paper is closely related to Bao et al. (2024a) and Weinstein & Ramdas (2020). To the best of our knowledge, the literature on the online selective conformal inference is limited to these. We provide a brief discussion of other work, which are only related to our paper in a broader sense.

Traditional selective inference provides valid statistical inference after data-driven selection (e.g. model selection). These methods condition on the selection event to correct for "cherry-picking" and avoid overstated significance. A prominent example is the exact post-selection inference framework for the Lasso by Lee et al. (2016). Such classical approaches, including Fithian et al. (2014), guarantee nominal Type I error given the selection, typically under parametric assumptions (e.g. Gaussian noise). However, they often require specialized derivations for each procedure and may be limited in scope (e.g. specific to linear models). Recently, this motivated *distribution-free* selective inference, where conformal prediction plays a key role. However, most of these works (discussed below) are in the offline setting.

A central idea is to create selection-aware conformal p-values or intervals that account for the fact that one only reports results on selected instances or hypotheses. Several recent works implement this idea. Jin & Candès (2023) introduced conformal p-values for identifying instances with large outcomes while controlling false discoveries. The method wraps around any predictive model to output p-values for each test instance that its outcome exceeds a specified threshold. By applying a Benjamini–Hochberg (BH) (Benjamini & Hochberg, 1995) procedure to these p-values, one can select a subset of candidates and control the false discovery rate (FDR) in finite samples.

Bao et al. (2024b) construct prediction intervals for selected individuals while controlling the FCR. They develop a split-conformal procedure called SCOP (Selective COnditional conformal Predictions) that uses a held-out calibration set to mimic the selection on the test set. By performing the same selection on calibration data and constructing conformal intervals on those subsets, SCOP achieves exact FCR control under exchangeability. Notably, it improves interval efficiency: whereas a naive FCR adjustment (Benjamini & Yekutieli, 2005) would inflate all intervals uniformly, SCOP yields narrower intervals in practice while still guaranteeing that a chosen proportion of the reported intervals covers the truth.

Wang et al. (2025) tackle selection effects in multiple hypothesis testing powered by conformal prediction. They introduce a selective conformal p-value that adjusts for a broad class of selection procedures (for instance, selecting top-scoring instances for testing). The key idea is to use a holdout set to simulate the selective distribution—effectively, recalibrating conformal p-values on data that underwent a similar selection rule. By adaptively choosing the calibration data based on the stability of the selection rule, they ensure the calibration set is exchangeable with the selected test point. This yields valid post-selection p-values which, when plugged into BH, control the FDR on the selected subset.

Another recent work, Bai & Jin (2024), addresses a practical challenge: how to perform model selection or tuning within a conformal inference procedure without invalidating the guarantees. Typically, if one uses the same data to choose a predictive model (or conformity score) and to calibrate prediction sets, the exchangeability needed for conformal validity can be broken. Existing solutions often demand data splitting or a fixed model choice independent of calibration data. They provide a general framework for valid post-selection inference even when models or scores are optimized on the data.

As mentioned earlier, all these above works are in the offline setting, and thus extending the above papers to the online setting seems a ripe direction for future work.

## 8 Discussion

Our findings highlight that the choice of calibration strategy is crucial in ensuring key inferential guarantees—namely, selection-conditional coverage and FCR control. The theoretical results presented in this paper reveal that existing calibration selection strategies, such as the recently proposed adaptive strategy (`ADA`), do not necessarily preserve the exchangeability between the calibration set and the selected test datum. As a result, they fail to provide valid selection-conditional coverage guarantees in general. In contrast, the proposed family of exchangeability-preserving calibration selection strategies provides selection-conditional coverage.

There are several avenues for future research. First, there appears to be a trade-off between strict theoretical guarantees (via exchangeability preservation) and practical efficiency (in terms of calibration set size and interval width). Investigating adaptive or hybrid selection strategies that balance these objectives (like `K-EXPRESS` or `EXPRESS-M`) is a promising direction. Second, while our analysis assumes decision driven selection rules, extending the framework to other (potentially more general) selection rules remains an important challenge. Third, empirical studies across diverse applications—such as medical decision-making and finance—could provide insights into the real-world applicability of these methods.

**Acknowledgments**

The authors thank Margaux Zaffran for helpful comments. Yusuf Sale is supported by the DAAD program Konrad Zuse Schools of Excellence in Artificial Intelligence, sponsored by the Federal Ministry of Education and Research.

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

# A  Preliminaries

For completeness, we provide basic results on quantile and related concepts, which are already well-documented in the conformal prediction literature (Vovk et al., 2005; Lei et al., 2018; Romano et al., 2019; Barber et al., 2023). For a comprehensive overview of the theoretical foundations of conformal prediction, see Angelopoulos et al. (2024a).

Let $Z$ be a random variable with cumulative distribution function (cdf) $F_Z$. The *quantile function* $Q(\alpha)$ is defined as the (generalized) inverse of $F_Z$, for $\alpha \in (0, 1)$. Formally,

$$Q(\alpha) = F_Z^{-1}(\alpha)$$

$$= \inf\{z \in \mathbb{R} : F_Z(z) \geq \alpha\}.$$

Equivalently, $Q(\alpha)$ is the smallest real number $z$ such that $F_Z(z) \geq \alpha$. Now suppose we have a sample $Z_1, \ldots, Z_n$ drawn from the distribution of $Z$. The *empirical* cdf $\widehat{F}_n$ is defined by

$$\widehat{F}_n(z) = \frac{1}{n} \sum_{i=1}^{n} \mathbb{1}\{Z_i \leq z\},$$

where $\mathbb{1}\{\cdot\}$ denotes the indicator function. Then the *empirical* quantile function $\widehat{Q}_n(\alpha)$ is given as the inverse of $\widehat{F}_n$:

$$\widehat{Q}_n(\alpha) = \inf\{z \in \mathbb{R} : \widehat{F}_n(z) \geq \alpha\}.$$

More explicitly, if we denote the order statistics by $Z_{(1)} \leq Z_{(2)} \leq \cdots \leq Z_{(n)}$, then

$$\widehat{Q}_n(\alpha) = Z_{(\lceil n\alpha \rceil)},$$

where $\lceil n\alpha \rceil$ is the ceiling of $n\alpha$.

**Lemma A.1** (Quantile lemma). *Let $Z_1, \ldots, Z_n$ be exchangeable random variables. Then, for any $\alpha \in (0, 1)$*

$$\mathbb{P}(Z_n \leq \widehat{Q}_n(\alpha)) \geq \alpha.$$

*Additionally, if the random variables $Z_1, \ldots, Z_n$ are almost surely distinct, then*

$$\mathbb{P}(Z_n \leq \widehat{Q}_n(\alpha)) \leq \alpha + \frac{1}{n}.$$

**Lemma A.2** (Quantile inflation). *Let $Z_1, \ldots, Z_{n+1}$ be exchangeable random variables. Then, for any $\alpha \in (0, 1)$*

$$\mathbb{P}(Z_n \leq \widehat{Q}_n((1 + 1/n)\alpha)) \geq \alpha.$$

*Additionally, if the random variables $Z_1, \ldots, Z_{n+1}$ are almost surely distinct, then*

$$\mathbb{P}(Z_n \leq \widehat{Q}_n((1 + 1/n)\alpha)) \leq \alpha + \frac{1}{n+1}.$$

## B  Omitted Proofs

The proof of Theorem 4.5 follows from Lemma 4.4 and classical conformal arguments (see in particular Angelopoulos et al. (2024a) and references therein).

**Lemma B.1.** *Let the selection procedure be instantiated with strategy (*iii*), (*v*) or (*vi*). Then, $\{Z_i\}_{i \in \widetilde{\mathcal{D}}_t}$ are exchangeable conditional on the event that $S_0 = s_0, \dots, S_{t-1} = s_{t-1}$, where $(s_0, \dots, s_{t-1}) \in \{0,1\}^{t-1}$.*

*Proof.* Let $(s_0, \dots, s_{t-1}) \in \{0,1\}^{t-1}$ and $D$ any set of indices. Further, denote by $\mathcal{E}$ the event that $\widetilde{\mathcal{I}}_t(Z_{-n}, \dots, Z_t) = D$ and $S_0 = s_0, \dots, S_{t-1} = s_{t-1}$, and assume $\mathbb{P}(\mathcal{E}) > 0$. For any measurable $A \subseteq (\mathcal{X} \times \mathcal{Y})^{|D|}$, let $B = \Big\{ (z_{-n}, \dots, z_t) \in (\mathcal{X} \times \mathcal{Y})^{N+1} : (z_i)_{i \in D} \in A,\ \widetilde{\mathcal{I}}_t(z_{-n}, \dots, z_t) = D,\ \mathcal{S}_j(x_j) = s_j,\ 0 \le j \le t-1 \Big\}$. Thus, $\{(Z_{-n}, \dots, Z_t) \in B\} = \{(Z_i)_{i \in D} \in A,\ \mathcal{E}\}$. By Lemma 4.4, both $\widetilde{\mathcal{I}}(\cdot)$ and $\mathcal{S}_j(\cdot)$ are permutation invariant, for $0 \le j \le t-1$. This yields $\{(Z_{-n}, \dots, Z_t) \in B\} = \{(Z_{\widetilde{\pi}(-n)}, \dots, Z_{\widetilde{\pi}(t)}) \in B\}$ and with that

$$\mathbb{P}\left((Z_i)_{i \in D} \in A,\ \mathcal{E}\right) = \mathbb{P}\left((Z_{-n}, \dots, Z_t) \in B\right)$$

$$= \mathbb{P}\left((Z_{\widetilde{\pi}(-n)}, \dots, Z_{\widetilde{\pi}(t)}) \in B\right)$$

$$= \mathbb{P}\left((Z_{\pi(i)})_{i \in D} \in A,\ \mathcal{E}\right),$$

where the second equality follows from exchangeability of $\{Z_i\}_{i=1}^t$. $\qquad\square$

**Lemma B.2.** *Let the selection procedure be instantiated with strategy (*iii*), (*v*) or (*vi*), and $(s_0, \dots, s_{t-1}) \in \{0,1\}^{t-1}$. If we have $\mathbb{P}(S_0 = s_0, \dots, S_{t-1} = s_{t-1}, S_t = 1) > 0$, then*

$$p_{\widetilde{\mathcal{D}}_t} = \frac{\sum_{j \in \widetilde{\mathcal{D}}_t} \mathbb{1}\{R_j \ge R_t\}}{|\widetilde{\mathcal{D}}_t|} \tag{16}$$

*is a valid p-value, i.e., $\mathbb{P}(p_{\widetilde{\mathcal{D}}_t} \le \alpha \mid S_0 = s_0, \dots, S_{t-1} = s_{t-1}, S_t = 1) \le \alpha$ for any $\alpha \in [0,1]$.*

*Proof.* Let $\mathcal{E}_t$ denote the event that $t \in \widetilde{\mathcal{D}}_t \equiv \widetilde{\mathcal{I}}_t(Z_{-n}, \dots, Z_t)$, which by (4) is equal to the event that $S_t = 1$, and $S_0 = s_0, \dots, S_{t-1} = s_{t-1}$, for $(s_0, \dots, s_{t-1}) \in \{0,1\}^{t-1}$. Conditional on $\mathcal{E}_t$, the non-conformity scores $\{R_i\}_{i \in \widetilde{\mathcal{D}}_t}$ are exchangeable. This implies that

$$p_{\widetilde{\mathcal{D}}_t} = \frac{\sum_{j \in \widetilde{\mathcal{D}}_t} \mathbb{1}\{R_j \ge R_t\}}{|\widetilde{\mathcal{D}}_t|} \tag{17}$$

*is a valid p-value, i.e., $\mathbb{P}\left(p_{\widetilde{\mathcal{D}}_t} \le \alpha \mid \mathcal{E}_t\right) \le \alpha$, implying that $\mathbb{P}\left(p_{\widetilde{\mathcal{D}}_t} \le \alpha \mid S_0 = s_0, \dots, S_{t-1} = s_{t-1}, S_t = 1\right) \le \alpha$.* $\qquad\square$

*Proof of Theorem 4.5.* Since we have $Y_t \in \widehat{C}_t(X_t) \iff p_{\widetilde{\mathcal{D}}_t} > \alpha$ by construction and classical conformal arguments, the strong selection-conditional coverage guarantee follows by Lemma B.2. $\qquad\square$

The proofs of Proposition 5.2 and Proposition 6.1 are adapted from Theorem 2 and Theorem 5 in Weinstein & Ramdas (2020), respectively, with modifications to align with the conformal framework.

*Proof of Proposition 5.2.* First, let $(s_0, \ldots, s_T) \in \{0, 1\}^{t-1}$ be any sequence, such that $\sum_i s_i > 0$. Now, assume that $\mathbb{P}\left\{Y_t \in \widehat{C}_t(X_t) \mid S_0 = s_0, \ldots, S_{t-1} = s_{t-1}, S_t = 1\right\} \geq 1 - \alpha$ for any $t \geq 0$. Then,

$$\mathbb{E}\left[\frac{\sum_{t=0}^T S_t \mathbb{1}\{Y_t \notin \widehat{C}_t(X_t)\}}{\sum_{j=0}^T S_j} \,\Big|\, S_0 = s_0, \ldots, S_T = s_T\right]$$

$$= \frac{1}{\sum_{j=0}^T s_j} \mathbb{E}\left[\sum_{t=0}^T S_t \mathbb{1}\{Y_t \notin \widehat{C}_t(X_t)\} \,\Big|\, S_0 = s_0, \ldots, S_T = s_T\right]$$

$$= \frac{1}{\sum_{j=0}^T s_j} \mathbb{E}\left[\sum_{t \leq T: s_t = 1} \mathbb{1}\{Y_t \notin \widehat{C}_t(X_t)\} \,\Big|\, S_0 = s_0, \ldots, S_T = s_T\right]$$

$$= \frac{1}{\sum_{j=0}^T s_j} \sum_{t \leq T: s_t = 1} \mathbb{P}\left[Y_t \notin \widehat{C}_t(X_t) \,\Big|\, S_0 = s_0, \ldots, S_T = s_T\right]$$

$$\overset{(a)}{=} \frac{1}{\sum_{j=0}^T s_j} \sum_{t \leq T: s_t = 1} \mathbb{P}\left[Y_t \notin \widehat{C}_t(X_t) \,\Big|\, S_0 = s_0, \ldots, S_{t-1} = s_{t-1}, S_t = 1\right]$$

$$\overset{(b)}{\leq} \frac{1}{\sum_{j=0}^T s_j} \sum_{t \leq T: s_t = 1} \alpha$$

$$= \alpha,$$

where $(a)$ follows since given $(S_0, \ldots, S_t)$ the selection decisions $(S_{t+1}, \ldots, S_T)$ are independent of $\{Y_t \notin \widehat{C}_t(X_t)\}$. Inequality $(b)$ follows from strong selection-conditional coverage. Now, taking the expectation over the conditional distribution of $S_1, \ldots, S_T$ given $\sum_i S_i > 0$ yields

$$\text{pFCR}(T) = \mathbb{E}\left[\frac{\sum_{t=0}^T S_t \mathbb{1}\{Y_t \notin \widehat{C}_t(X_t; \alpha_t)\}}{\sum_{j=0}^T S_j} \,\Big|\, \sum_i S_i > 0\right] \leq \alpha.$$

With that

$$\text{FCR}(T) = \mathbb{E}\left[\frac{\sum_{t=0}^T S_t \mathbb{1}\{Y_t \notin \widehat{C}_t(X_t; \alpha_t)\}}{\sum_{j=0}^T S_j} \,\Big|\, \sum_i S_i > 0\right] \cdot \mathbb{P}\left(\sum_i S_i > 0\right)$$

$$\leq \mathbb{E}\left[\frac{\sum_{t=0}^T S_t \mathbb{1}\{Y_t \notin \widehat{C}_t(X_t; \alpha_t)\}}{\sum_{j=0}^T S_j} \,\Big|\, \sum_i S_i > 0\right]$$

$$\leq \alpha.$$

Thus, $\text{FCR}(T) \leq \alpha$. Clearly, if $(a)$ holds with equality, then $\text{pFCR}(T) = \alpha$. Additionally, if $\mathbb{P}(\sum_i S_i > 0) = 1$, then $\text{FCR}(T) = \alpha$. This completes the proof. $\qquad\square$

*Proof of Proposition 6.1.* Let $T \geq 0$ and $\{S_j^{(t)}\}_{j\geq0}$ be a sequence of selection decisions $\{S_j\}_{j\geq0}$, where we set $S_t = 1$ deterministically. Now, define $\mathcal{F}^{T\backslash t} = \sigma\left(S_1, \ldots, S_{t-1}, S_{t+1}, \ldots, S_T\right)$. Then, we have

$$
\begin{aligned}
\mathrm{FCR}(T) &= \sum_{t=0}^{T} \mathbb{E}\left[\frac{S_t \mathbb{1}\{Y_t \notin \widehat{C}_t(X_t; \alpha_t)\}}{1 \vee \sum_{j=0}^{T} S_j}\right] \\
&= \sum_{t=0}^{T} \mathbb{E}\left[\frac{S_t \mathbb{1}\{Y_t \notin \widehat{C}_t(X_t; \alpha_t)\}}{\sum_{j=0}^{T} S_j^{(t)}}\right] \\
&\stackrel{(a)}{\leq} \sum_{t=0}^{T} \mathbb{E}\left[\frac{\mathbb{1}\{Y_t \notin \widehat{C}_t(X_t; \alpha_t)\}}{\sum_{j=0}^{T} S_j^{(t)}}\right] \\
&= \sum_{t=0}^{T} \mathbb{E}\left[\frac{1}{\sum_{j=0}^{T} S_j^{(t)}} \mathbb{E}\left[\mathbb{1}\{Y_t \notin \widehat{C}_t(X_t; \alpha_t)\} \mid \mathcal{F}^{T\backslash t}\right]\right] \\
&= \sum_{t=0}^{T} \mathbb{E}\left[\frac{1}{\sum_{j=0}^{T} S_j^{(t)}} \mathbb{E}\left[\mathbb{1}\{R_t > \widehat{Q}_{1-\alpha_t}(\{R_i\}_{i\in\mathcal{D}_t})\} \mid \mathcal{F}^{T\backslash t}\right]\right] \\
&\stackrel{(b)}{\leq} \sum_{t=0}^{T} \mathbb{E}\left[\frac{\alpha_t}{\sum_{j=0}^{T} S_j^{(t)}}\right] \\
&\stackrel{(c)}{\leq} \sum_{t=0}^{T} \mathbb{E}\left[\frac{\alpha_t}{\sum_{j=0}^{T} S_j}\right] \\
&\stackrel{(d)}{\leq} \alpha,
\end{aligned}
$$

Here $(a)$ follows since $S_t \leq 1$, and $(b)$ since $\alpha_t$ is $\mathcal{F}_{t-1}$-measurable, we have for a fixed $\alpha_t \in (0,1)$ that $\mathbb{E}\left[\mathbb{1}\{R_t > \widehat{Q}_{1-\alpha_t}(\{R_i\}_{i\in\mathcal{D}_t})\} \mid \mathcal{F}^{T\backslash t}\right] = \mathbb{E}\left[\mathbb{1}\{R_t > \widehat{Q}_{1-\alpha_t}(\{R_i\}_{i\in\mathcal{D}_t})\}\right]$. Finally, $(c)$ is due to the monotonicity of the selection rules and the last inequality $(d)$ follows from the invariant (14). $\qquad\square$

## C  Other Conformal Methods

### C.1  LORD-CI

For completeness, we provide a brief overview of the LORD-CI algorithm in this section, slightly altered from its original presentation for confidence sets to fit the conformal framework. The algorithm was initially proposed by Weinstein & Ramdas (2020), to which we refer for a more comprehensive treatment. Let $\alpha_t \in (0,1)$ be a $\mathcal{F}_t$-measurable coverage level, then a conformal prediction interval is constructed as

$$\widehat{C}_t(X_t) = \{y \in \mathcal{Y} : R(X_t, y) \leq \widehat{Q}_{1-\alpha_t}(\{R_i\}_{i \in \mathcal{J}_{t,\text{off}}})\}.$$

The marginal level $\alpha_t$ is updated dynamically by maintaining the *invariant*

$$\frac{\sum_{t=0}^{T} \alpha_t}{1 \vee \sum_{t=0}^{T} S_t} \leq \alpha, \quad \text{for any } T \geq 0.$$

Algorithm 1 is an explicit instantiation of LORD-CI, which maintains the above invariant. Initially, before any selection, the algorithm only has the small budget $W_0 \in (0, \alpha)$ to spend (spread out over time by $\gamma_t$). Once the first interval is reported, the algorithm gains an additional "error wealth" of size $(\alpha - W_0)$. After the first selection, any future time point $t$ gets an increment of $(\alpha - W_0)\gamma_{\tau_1}$ in its threshold. Similarly, every subsequent selection contributes an additional $\alpha$ distributed over future times. Thus, by time $t$, $\alpha_t$ is the sum of contributions from the initial budget and all past selections' budgets allocated to time $t$. This mechanism guarantees that $\alpha_t$ is non-decreasing over time — whenever a new selection occurs, future $\alpha$-levels increase or stay the same.

---

**Algorithm 1** LORD-CI procedure (Weinstein & Ramdas, 2020, modified)

---

1: **Input:** offline data $\mathcal{J}_{t,\text{off}}$; sequence $\{Z_i\}$ observed sequentially; deterministic sequence $\{\gamma_i\}$ summing to one; constant $W_0 \in (0, \alpha)$; selection rules $\{\mathcal{S}_i\}$; $\alpha \in (0,1)$
2: **Output:** online FCR-adjusted selective prediction intervals
3: $t \leftarrow 1$
4: **for** $j = 1, 2, \dots$ **do**
5:     **while** $\mathcal{S}_t(X_t) = 0$ **do**
6:         $t \leftarrow t + 1$
7:     **end while**
8:     $\tau_j \leftarrow t$
9:     $\alpha_t \leftarrow \gamma_t W_0 + (\alpha - W_0)\gamma_{t-\tau_1} + \alpha \sum_{\{k:\tau_k < t, \tau_k \neq \tau_1\}} \gamma_{t-\tau_k}$          ▷ Monotone update rule
10:     Report $\widehat{C}_t(X_t) = \{y \in \mathcal{Y} : R(X_t, y) \leq \widehat{Q}_{1-\alpha_t}(\{R_i\}_{i \in \mathcal{J}_{t,\text{off}}})\}$.
11:     $t \leftarrow t + 1$
12: **end for**

---

### C.2  Adaptive Conformal Inference

The ACI algorithm (Gibbs & Candes, 2021) adjusts the miscoverage level based on historical under- or over-coverage feedback. Specifically, for a target miscoverage level $\alpha$, the algorithm updates $\alpha_t$ according to

$$\alpha_t = \alpha_{t-1} + \gamma \left( \alpha - \mathbb{1}\{Y_{t-1} \notin \widehat{C}_t(X_{t-1}, \alpha_{t-1})\} \right),$$

where $\gamma > 0$ is a fixed step-size parameter. Here, $\widehat{C}_t(X_{t-1}, \alpha_{t-1})$ denotes the prediction interval constructed at time $t-1$ with nominal miscoverage level $\alpha_{t-1}$. We note that, the performance of ACI heavily relies on a well-specified step-size parameter. Gibbs & Candes (2021) recommend that the step-size should be chosen proportionally to the underlying rate of change in the environment, which is unknown in practice.

# D    Additional Simulations

We provide additional simulation results to complement those presented in the main part of the paper. First, we recall the following (decision driven) selection rules:

| Selection rule (A) | Selection rule (B) | Selection rule (C) |
|---|---|---|
| $x \mapsto \begin{cases} \mathbb{1}\left\{ x < \frac{1}{\tau_0} \sum_{i=0}^{j-1} S_i \right\} \text{ for } j \leq t-1 \\ \mathbb{1}\left\{ \sum_{i=0}^{j-1} S_i > \tau_1 \right\} \quad \text{ for } j = t, \end{cases}$ | $x \mapsto \mathbb{1}\left\{ x < \tau_1 + \frac{1}{\tau_0} \sum_{i=0}^{t-1} S_i \right\}$ | $x \mapsto \mathbb{1}\left\{ x > \tau_1 - \min\left( \frac{1}{\tau_0} \sum_{i=0}^{t-1} S_i, 2 \right) \right\}$ |

We evaluate several calibration selection strategies for online selective conformal prediction, which were discussed earlier. The aim is to generate prediction intervals for a test point based on previously selected calibration points, while ensuring valid selection-conditional coverage. We conduct Monte Carlo simulations to accumulate metrics, such as miscoverage and calibration set size. In each iteration of the simulation, a data set of size $N = N_{\text{off}} + N_{\text{on}}$, where $N_{\text{off}} = |\mathcal{J}_{t,\text{off}}|$ and $N_{\text{on}} = |\mathcal{J}_{t,\text{on}}|$ is generated. The generated data serve as potential $t$-calibration data.

In the simulations that follow, we generate $X \sim \text{Unif}[0, 2]$, and $Y = \mu(X) + \varepsilon$, where $\mu(X) = X\beta$. Additionally, we assume $\varepsilon \mid X \sim \mathcal{N}(0, X/2)$.

## D.1    Selection-Conditional Coverage

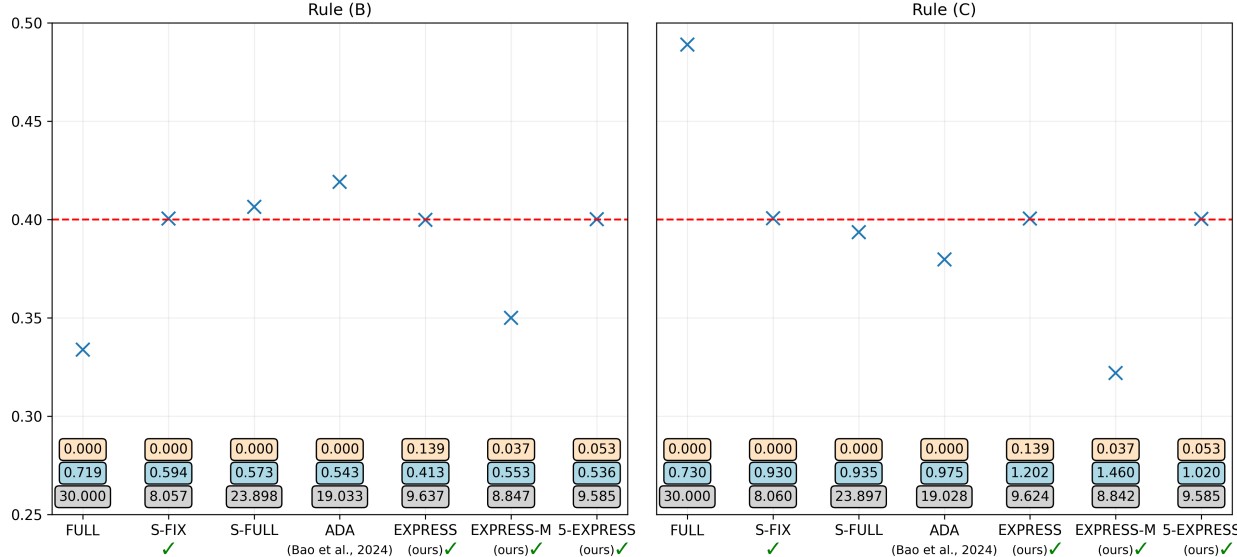

Figure 5: Miscoverage ($\times$) is shown alongside the number of calibration points (■), median interval length (■) and the fraction of infinite length prediction intervals (■). We highlight provably correct methods (✔) and the target level (- -). *All metrics are averaged over $N = 1 \times 10^6$ runs.*

**Simulation D.1.** We set $N_{\text{off}} = 10$ and $N_{\text{on}} = 20$. We perform two separate simulations, one using rule (B) and the other using rule (C). We then perform online selective conformal prediction with both existing and novel strategies. All reported metrics are averaged over $N = 1 \times 10^6$ runs.

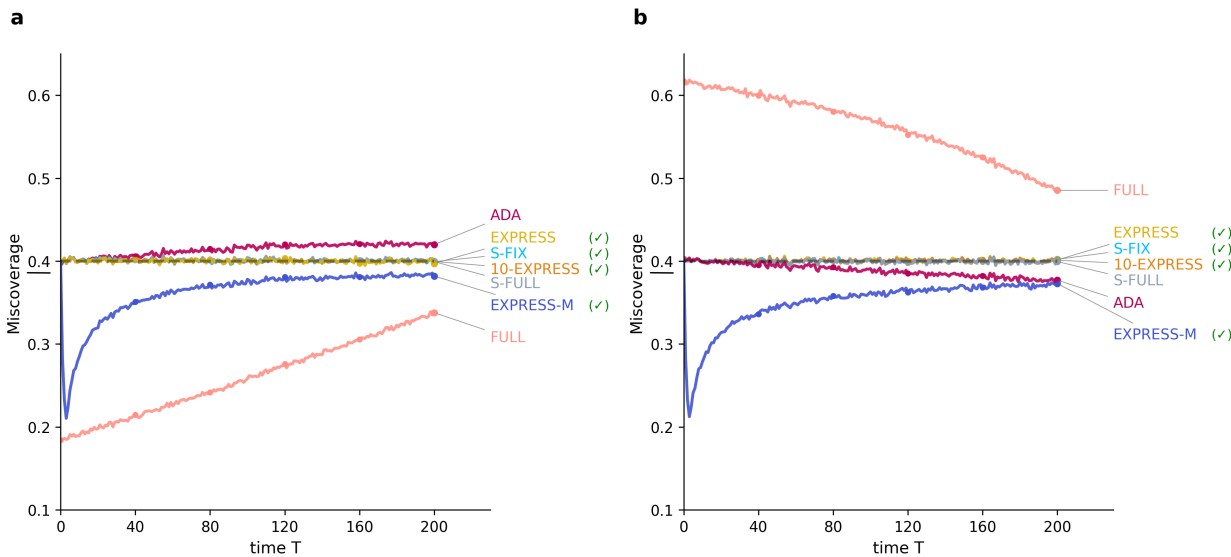

Figure 6: Miscoverage over time. The dashed black line represents the target level $\alpha = 0.4$. We highlight provably correct methods (✔). **(a)** Miscoverage for selection rule (B). **(b)** Miscoverage for selection rule (C). *Averaged over $N = 1 \times 10^4$ runs.*

**Simulation D.2.** We set $N_{\text{off}} = 50$ and $N_{\text{on}} = 200$. We run two separate simulations, one using rule (B) and the other using rule (C). We then perform online selective conformal prediction with both existing and novel strategies. At each time $T \geq 0$ we report the miscoverage, averaged over $N = 1 \times 10^4$ runs.

### D.2 FCR Control

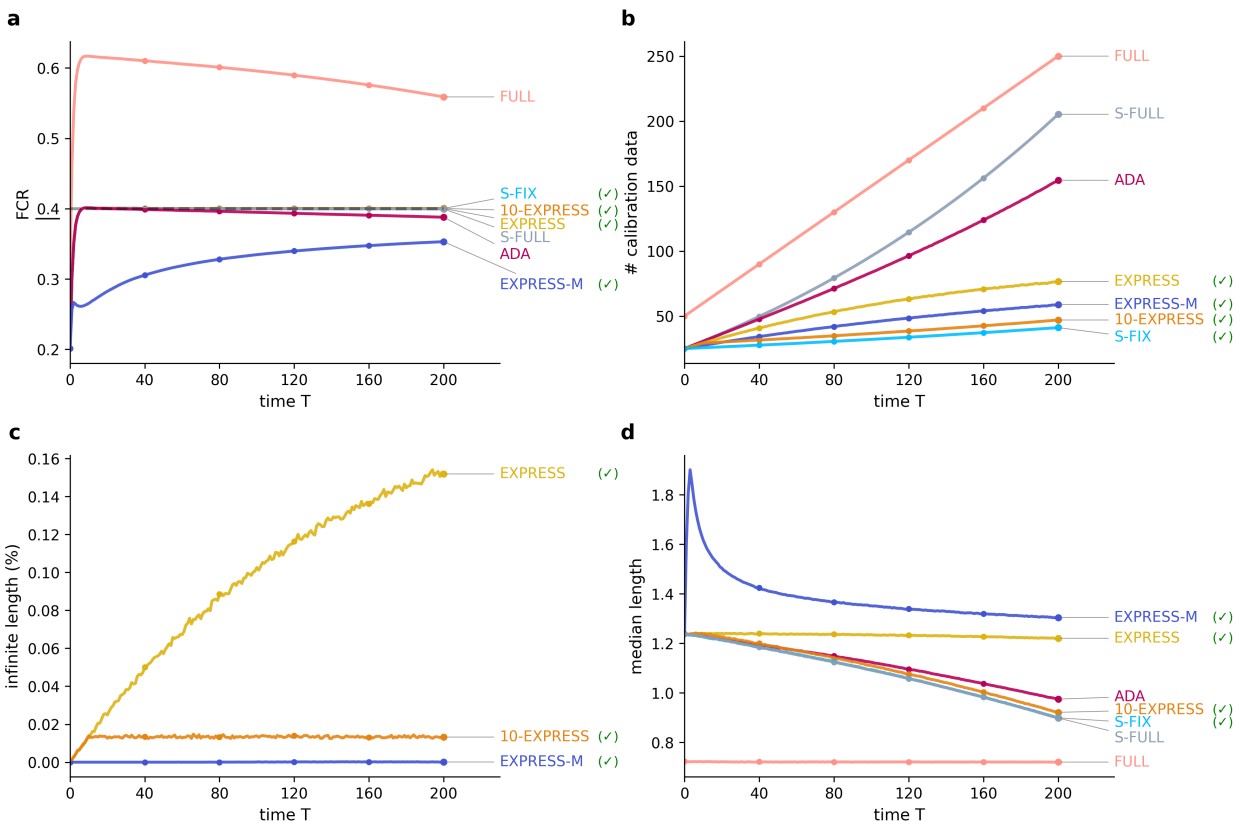

Figure 7: **(a)** FCR as a function of time $T$. The dashed black line represents the target level $\alpha = 0.4$. We highlight provably correct methods (✔). **(b)** Number of calibration data points used over time. Strategies accumulating more calibration data tend to yield shorter prediction intervals. **(c)** Fraction of prediction intervals of infinite length over time. A high fraction suggests a strategy often fails to provide informative intervals. Only reported for novel strategies. **(d)** Median prediction interval length over time. Shorter intervals indicate higher informativeness. *All metrics are averaged over $N = 1 \times 10^4$ runs.*

**Simulation D.3.** We set $N_{\text{off}} = 50$ and $N_{\text{on}} = 200$. Here, we use selection rule (C). We then perform online selective conformal prediction with both existing and novel strategies.

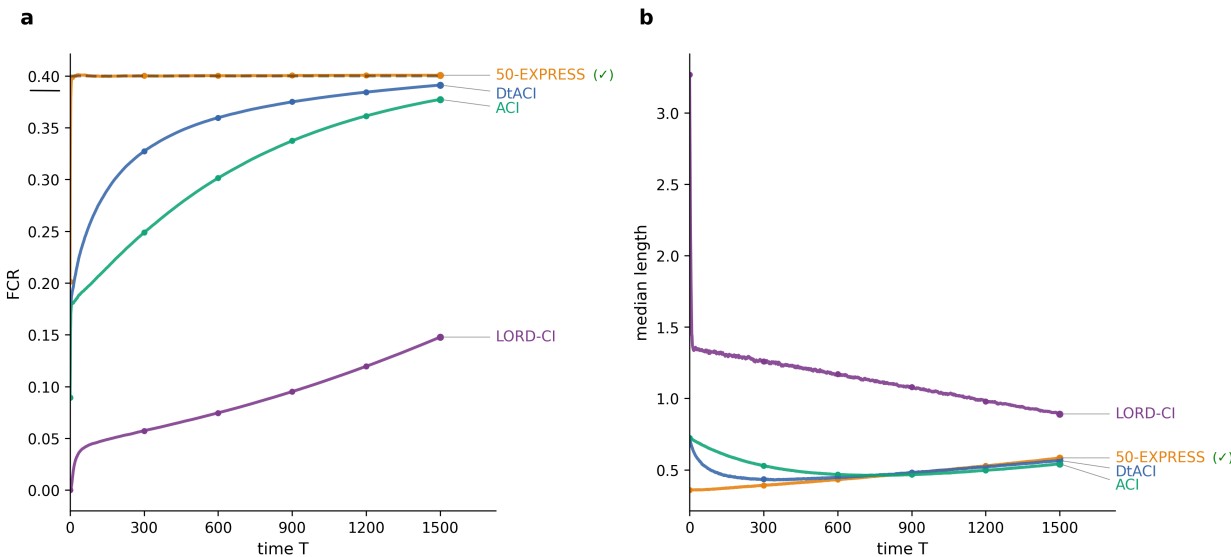

Figure 8: **(a)** FCR as a function of time $T$. The dashed black line represents the target level $\alpha = 0.4$. We highlight provably correct methods (✔). **(b)** Median prediction interval length over time. Shorter intervals indicate higher informativeness. *All metrics are averaged over $N = 1 \times 10^4$ runs.*

**Simulation D.4.** We set $N_{\text{off}} = 200$ and $N_{\text{on}} = 1500$. Here, we use selection rule (C). We compare online selective conformal prediction with 50-EXPRESS strategy, ACI algorithms and LORD-CI.

