# OpenReview forum: "Online Selective Conformal Inference: Errors and Solutions"
_TMLR — Accepted by TMLR_

### Review · Reviewer_rVzn · 2025-04-17

**Summary Of Contributions:**

This is a work on conformal prediction in which the authors review a few different calibration selection strategies in the setting where data arrives in an online fashion and for which there is a requirement that

$$
\forall t\geq 0:\ P(Y_t\in \hat{C}_t(X_t)|S_t=1)\geq 1-\alpha,
$$

where $S_t$ is a particular selection rule and $\hat{C}$ are the prediction intervals.

The authors show that three particular calibration strategies either violate exchangeability or are too stringent in doing so in the setting discussed. The authors also propose a series of calibration strategies, called EXPRESS, K-EXPRESS, and EXPRESS-M (“M” for “merge”, as the authors merge intervals from EXPRESS and S-FIX), that have varying degrees of stringiness and performance in terms of number of calibration points needed, size of intervals, percentage of unbounded intervals, etc.

All methods are compared in terms of synthetic toy experiments that showcase various metrics, e.g., miscoverage, interval length, etc. The authors provide theoretical results for when some strategies do not satisfy symmetry, as well as when they do (including guarantees on the false coverage rate).

**Audience:**

Yes

**Broader Impact Concerns:**

NA.

**Claims And Evidence:**

Yes

**Requested Changes:**

## Requested Changes

Some minor suggestions to strengthen presentation; they are not necessary for recommendation.

- Perhaps because of inexperience with the online setting, I didn’t immediately understand where $S_i$ come from in the synthetic examples (examples of selection rules are provided as a function of the selections $S_i$, but how selections are constructed, e.g., in the experiments, is left implicit for the reader at the moment). This may be standard in conformal prediction, but a sentence or two on may help readers.
- Typo: Above Equation (8), “Consequently, *display* (7)” → “Consequently, Equation (7)”.
- Although I enjoyed the miscoverage plots, I wonder if there's a more visual way to compare the methods, e.g., a parallel coordinates plot.

**Strengths And Weaknesses:**

## Strengths

- The paper is well-written and, with the addition of the information in the appendix, relatively easy to follow. There is a clear presentation of key results.
- The setup of the synthetic examples is simple enough for an interested reader to reproduce.
- Good theoretical results tying intricacies of the various calibration strategies with coverage guarantees, with a good mix of positive and negative results.
- The buildup from EXPRESS, to K-EXPRESS, and then EXPRESS-M helps to understand what each method brings to the table (which can also be seen in the toy examples).


## Weaknesses

- I think readers would have appreciated a realistic example  (like in the work by Bao[^1], et al., 2024) to highlight any other difficulties that are not seen in the synthetic examples.

[^1]: Bao, Y., Huo, Y., Ren, H. and Zou, C., 2024. Cap: A general algorithm for online selective conformal prediction with fcr control. _arXiv preprint arXiv:2403.07728_.

---

> ### Author Response · Authors · 2025-04-29
> **Response to reviewer rVzn**
>
> We would like to thank the reviewer for the careful reading of our paper, the very encouraging assessment, and the concrete suggestions for improvement. In the following we answer to each point individually.
>
> > I think readers would have appreciated a realistic example (like in the work by Bao[1], et al., 2024) to highlight any other difficulties that are not seen in the synthetic examples.
>
> Our goal in this paper was to scrutinize existing methods as transparently as possible by focusing on intuitive, synthetic (counter)-examples. This allowed us to isolate key behaviors and demonstrate proof‑of‑concept for our own proposals without confounding factors. Given that particular focus, we were not sure what a real data example would add to our paper.
>
> > Perhaps because of inexperience with the online setting, I didn’t immediately understand where $S_i$ come from in the synthetic examples (examples of selection rules are provided as a function of the selections, but how selections $S_i$ are constructed, e.g., in the experiments, is left implicit for the reader at the moment). This may be standard in conformal prediction, but a sentence or two on may help readers.
>
> We are happy to elaborate more on the selection *decisions* $\lbrace S_t \rbrace_{t \ge 0}$. For instance, we consider the following example: at each time $t \geq 0$ we form selection decisions $S_t$ by applying the (online) selection *rule*
>
> $$
> x \mapsto 1 \lbrace x < \tau_1 + \frac{1}{\tau_0} \sum_{i=0}^{t-1} S_i \rbrace.
> $$
>
> which is exactly the rule used in **Simulation 5.1** (pg. 10 ff.) and **Simulation 6.1** (pg. 12 ff.) For illustration purposes, let us further assume $\tau_0 = \tau_1 = 1$; the rule then simplifies to
>
> $$
> x \mapsto 1\lbrace x < 1 + \sum_{i=0}^{t-1} S_i \rbrace.
> $$
>
> The sequence of selection decisions $\lbrace S_t \rbrace_{t\geq0}$ is obtained recursively:
> - $t = 0$: the threshold equals $1$, so $S_0 = 1$ if $X_0 < 1$ and $S_0 = 0$ otherwise.
> - $t = 1$: the threshold is now $1 + S_0$; we set $S_1 = 1$ if $X_1 < 1 + S_0$ and $S_1 = 0$ otherwise.
> - $\dots$
>
> All other experiments follow the same principle; only the functional form of the threshold (or the constants $\tau_0, \tau_1$) changes. Following your suggestion, we have expanded on this point in the revised version of our paper.
>
> > Typo: Above Equation (8), “Consequently, display (7)” → “Consequently, Equation (7)”.
>
> Thank you for pointing this out. We will streamline correspondingly.
>
> > Although I enjoyed the miscoverage plots, I wonder if there's a more visual way to compare the methods, e.g., a parallel coordinates plot.
>
> We thank the reviewer for this suggestion. If the reviewer could elaborate on what kind of additional plot they are imagining, we are happy to include it.

---

> > ### Comment · Reviewer_rVzn · 2025-05-05
> > **Response to authors**
> >
> > Thank you for the comments.
> >
> > Re: including a realistic example, I understand the rationale of the authors; no comments there.
> >
> > Re: selection decisions, I appreciate the explanation, that seems clear to me. I didn't see an updated PDF in the revisions, maybe I missed it?
> >
> > Re: plots, I was just thinking about whether something like a [parallel coordinate plot](https://en.wikipedia.org/wiki/Parallel_coordinates) would make it a bit easier to compare the methods in Figure 2, although I've reviewed again the plot and there may be too many methods for this to be a useful visual.

---

> > > ### Author Response · Authors · 2025-05-22
> > > **Response to reviewer rVzn**
> > >
> > > We are happy that we could clarify the reviewer’s question and would like to thank them again for their thoughtful feedback.
> > >
> > > We also appreciate their patience: following the TMLR guidelines, we waited for the third review before uploading a revised version of the manuscript. The expanded explanation regarding selection decisions can now be found on *p. 5*, directly following *Definition 2.1*.
> > >
> > > If there is anything else that remains unclear or if the reviewer has additional questions, we would be glad to elaborate further.

---

> > > > ### Comment · Reviewer_rVzn · 2025-05-26
> > > > **Response to authors**
> > > >
> > > > > We also appreciate their patience: following the TMLR guidelines, we waited for the third review before uploading a revised version of the manuscript. The expanded explanation regarding selection decisions can now be found on p. 5, directly following Definition 2.1.
> > > > > If there is anything else that remains unclear or if the reviewer has additional questions, we would be glad to elaborate further.
> > > >
> > > >
> > > > Sounds good, I don't have any further questions. Thank you for responding promptly.

---

### Review · Reviewer_8mFZ · 2025-05-05

**Summary Of Contributions:**

This paper considers the problem of online conformal prediction under the online selection rule. The prediction intervals need to be correct for the selected data, while the online selection may break the exchangeability between the test datum and the rest. This paper points out that the exchangeability is actually violated in existing strategies and proposes a sufficient condition for exchangeability. The authors also provide an algorithm (EXPRESS) and its variants that satisfy this condition and verify its effectiveness via simulation.

**Audience:**

Yes

**Broader Impact Concerns:**

None.

**Claims And Evidence:**

Yes

**Requested Changes:**

1. Can you provide some more discussions on Assumption A? I don't quite see why it doesn't hurt the applicability of your algorithms. What if the assumption is violated? Is this assumption necessary for online selective conformal prediction?

2. Can you provide some more numerical experiments? For example, on real data, or based on some other selection rules than (11).

3. Can you provide a little more explanation on EXPRESS-M at the end of Section 3? I'm still confused about how you merge S-FIX and EXPRESS. The fractions don't add up to 1.

**Strengths And Weaknesses:**

Strengths:

1. This paper clearly identifies one sufficient (yet not necessary) condition for the exchangeability to hold under online selective conformal prediction. It also points out the missing part ignored by current works.

2. This paper proposes one algorithm (EXPRESS) and two practical variants that achieve the exchangeability.

3. The authors show the connection to a broader family of online conformal prediction algorithms in Section 6.

Weaknesses:

1. The assumption A is too strong. Only considering the decision rules purely dependent on the online data limits the applicability.

2. The sufficient condition for exchangeability is too restrictive. The practical value of the proposed algorithms is somewhat limited due to a non-zero fraction of infinite-length prediction intervals. The simulation results only focus on the median of the length rather than the mean of the length, further verifying that point.

3. The empirical evidence is purely synthetic.

---

> ### Author Response · Authors · 2025-05-22
> **Response to reviewer 8mFZ**
>
> We sincerely thank the reviewer for their thoughtful and encouraging feedback. We’re happy that our work’s contributions and relevance were recognized. In the following we answer to each point individually.
>
> >The assumption A is too strong. Only considering the decision rules purely dependent on the online data limits the applicability.
>
> We note critically that Bao et al. (2024a) operate under the same Assumption (A), namely that decision-driven selection rules are independent of the offline data. Since our paper aims to pinpoint fundamental errors in their proposal, we adopt this assumption as well to mirror their setting and enable a direct, transparent comparison between existing and novel calibration selection strategies.
>
> However, if one were to select calibration points from the offline data when Assumption (A) does \emph{not} hold, then Lemma 4.2 might be violated as well. A straightforward workaround is to avoid offline calibration altogether and base all calibration selection solely on the online stream. While this avoids reliance on Assumption (A), it comes at the cost of (potentially) reduced calibration set size, as the offline pool is no longer used for calibration selection.
>
> We admit that the mildness of the assumption is subjective, and future work can attempt to eliminate this assumption, but this does not appear easy to us (and to Bao et al., which is the prime precursor to our work).
>
> > The sufficient condition for exchangeability is too restrictive. The practical value of the proposed algorithms is somewhat limited due to a non-zero fraction of infinite-length prediction intervals. The simulation results only focus on the median of the length rather than the mean of the length, further verifying that point.
>
> We thank the reviewer for highlighting the trade-off inherent in enforcing an exchangeability-preserving strategy. We ourselves were transparent about this, and explicitly noted in our manuscript (p. 5) that **EXPRESS** indeed enforces a stringent symmetry condition and consequently produces a non-trivial fraction of infinite-length intervals. To address precisely this concern, we developed two practical variants:
>
> - **K-EXPRESS**, which only enforces exchangeability over the last $k$ decision points, and
> - **EXPRESS-M**, which merges **S-FIX** and **EXPRESS** intervals via a Bonferroni split of $\alpha \in (0,1)$.
>
>
> As shown in Figure 3(c) (p. 10), both **K-EXPRESS** and **EXPRESS-M** reduce the fraction of infinite intervals to a negligible level, while still preserving exact selection-conditional coverage and controlling FCR, hence offering a more practical solution.
>
> We emphasize that infinite-length intervals are a well-known pain point in online conformal methods, such as adaptive conformal inference (ACI) (Gibbs and Candès, 2021), where they arise frequently in practice. This issue has also been explicitly highlighted recently by  Angelopoulos (2023).
>
> > The empirical evidence is purely synthetic.
>
> Our goal in this paper was to scrutinize past work as transparently as possible by focusing on intuitive, synthetic (counter)-examples, to clearly demonstrate their incorrect theoretical claims. This allowed us to isolate key behaviors and demonstrate proof‑of‑concept for our own proposals without confounding factors. Given that particular focus, we were not sure what a real data example would add to our paper.
>
> > Can you provide some more discussions on Assumption A? I don't quite see why it doesn't hurt the applicability of your algorithms. What if the assumption is violated? Is this assumption necessary for online selective conformal prediction?
>
> We expanded the discussion in Section 2 (p. 4) of the revised manuscript as follows (see also our earlier answers):
>
> *(1) Why did we introduce Assumption (A)?*
>
>  We adopted Assumption (A) to align with the setting of Bao et al. (2024a), who also operate under this assumption. This enables a direct comparison of existing and novel calibration selection strategies.
>
> *(2) What happens if Assumption (A) is violated?*
>
>  If Assumption (A) is violated, post-selection exchangeability may no longer hold, and the formal validity guarantees presented in this paper do not apply in general. However, as discussed earlier, one can circumvent the reliance on Assumption (A) by selecting calibration points exclusively from the online stream, thereby ensuring that the selection rules remain unaffected by the offline data.
>
> *(3) Is Assumption (A) necessary in general?*
>
>  Assumption (A) is necessary whenever one wishes to include offline data in the calibration selection procedure. Without it, the selection rules may implicitly depend on the offline data, breaking the permutation invariance required for post-selection exchangeability.

---

> ### Author Response · Authors · 2025-05-22
> **Response to reviewer 8mFZ**
>
> > Can you provide some more numerical experiments? For example, on real data, or based on some other selection rules than (11).
>
> We appreciate the suggestion to broaden our empirical evaluation. In fact, we have already included two further decision-driven selection rules, denoted (B) and (C) in Appendix D, and show in Figures 5-6 that all of our strategies continue to preserve selection-conditional coverage and control FCR under these markedly different rules. We will call this out explicitly in the main text.
>
> It is not clear to us what additional value a real data experiment will bring, but are happy to consider it if the reviewer has a compelling argument.
>
> > Can you provide a little more explanation on EXPRESS-M at the end of Section 3? I'm still confused about how you merge S-FIX and EXPRESS. The fractions don't add up to 1.
>
> We are happy to elaborate more on this point. **EXPRESS-M** simply intersects two prediction intervals, one from **S-FIX** at a smaller nominal level and one from **EXPRESS** at the complementary level, so that their miscoverage budgets sum exactly to $\alpha$. Let us illustrate this in the following:
>
> Suppose  $t > 0$, we then set
> \begin{align*}
> \alpha^{\text{SF}}  =  \frac{1}{\sqrt{t}} \alpha, \quad \alpha^{\text{EX}} =  \Bigl(1-\tfrac{1}{\sqrt{t}}\Bigr) \alpha.
> \end{align*}
>
> Note that $\alpha^{\text{SF}}  + \alpha^{\text{EX}}  = \alpha.$
>
> We then form
>
> \begin{align*}
> \widehat{C}^{\text{SF}}_t  =  \text{S-FIX interval at level } 1- \alpha^{\text{SF}}, \quad \widehat{C}^{\text{EX}}_t = \text{EXPRESS interval at level }1-\alpha^{\text{EX}},
> \end{align*}
> and take their intersection
> \begin{align*}
> \widehat{C}^{\text{M}}_t  =  \widehat{C}^{\text{SF}}_t  \cap  \widehat{C}^{\text{EX}}_t.
> \end{align*}
>
> By the union bound,
> \begin{align*}
> \mathbb{P} (Y_t\notin \widehat{C}^{\text{M}}_t | S_t = 1) \leq \mathbb{P} (Y_t\notin \widehat{C}^{\text{SF}}_t | S_t = 1 )  +  \mathbb{P}(Y_t\notin \widehat{C}^{\text{EX}}_t | S_t = 1)  \leq \alpha^{\text{SF}} + \alpha^{\text{EX}}  = \alpha
> \end{align*}
> guaranteeing overall miscoverage $\alpha$.
>
> We clarified on this, and explicitly noted that $\alpha^{\text{SF}}  + \alpha^{\text{EX}}  = \alpha$, in the revised manuscript (see p. 6)
>
>
> **References**
>
> Gibbs, I., \& Candes, E. (2021). Adaptive conformal inference under distribution shift. Advances in Neural Information Processing Systems, 34, 1660-1672.
>
> Angelopoulos, A., Candes, E., \& Tibshirani, R. J. (2023). Conformal pid control for time series prediction. Advances in neural information processing systems, 36, 23047-23074.

---

### Review · Reviewer_EKuR · 2025-05-21

**Summary Of Contributions:**

The paper studies online conformal inference in settings where there is a selection process---i.e., predictions are only given for selected time steps. This selection process can break the exchangeability property that is crucial to conformal prediction and to ensure that prediction intervals are valid conditional on the selection process.

The work studies how to select calibration data be chosen to ensure exchangeability and they discuss/evaluate six main calibration selection strategies.

**Audience:**

Yes

**Claims And Evidence:**

Yes

**Requested Changes:**

N/A

**Strengths And Weaknesses:**

Strengths:
- This is a strong paper from a theoretical point of view. The results are strong and extensive. The authors carefully study the properties of 6 selection rules in online conformal prediction, highlighting which of them guarantee "symmetry" (a property they show is crucial to the problem at hand) and hence lead to selection-conditional coverage.
- The message is clean: many of the existing selection techniques do not provide the desired coverage guarantees. In turn, part of the contribution of this work is also to introduce and develop novel selection strategies, on top of proving their properties.
- There are additional results in Section 5 that also study when a weaker notion of FCR control is satisfied, so the picture given by the paper is fairly complete.
- This is, to me, an especially important paper to accept because it corrects misconceptions and mistakes from previous work, and in that sense should prove to be an impactful paper to the conformal prediction community.

Weakness:
- This is mostly a theory paper so I think this is not too much of an issue here, since the experiments seem to mostly be illustrative, but I did find the experimental results weak in places. The setups of simulations 4.1 and 5.1 feels very specific and a bit arbitrary.

To me, this is a clear accept. I do want to note that my review is medium confidence, as, while I am familiar with conformal prediction, this is not my primary area of research.

---

> ### Comment · Reviewer_EKuR · 2025-05-21
>
> Also, dear authors, I deeply apologize for the delay in my review, due to an extremely busy end of semester...

---

> ### Author Response · Authors · 2025-05-22
> **Response to reviewer EKuR**
>
> We sincerely thank the reviewer for taking the time to carefully read and review our paper. We are very pleased to see that the theoretical contributions and overall message of the work were appreciated, and we are grateful for the recognition of its potential impact on the conformal prediction community.
>
> > This is mostly a theory paper so I think this is not too much of an issue here, since the experiments seem to mostly be illustrative, but I did find the experimental results weak in places. The setups of simulations 4.1 and 5.1 feels very specific and a bit arbitrary.
>
> We are happy to elaborate on this point. The aim of our simulations was to construct simple and transparent setups that isolate and clearly reveal the limitations of existing methods. In particular, Simulations 4.1 and 5.1 were deliberately designed as minimal counterexamples to demonstrate the failure of selection-conditional coverage and FCR control, respectively, despite prior claims suggesting these guarantees hold.
>
> Should any aspect remain unclear or if the reviewer has further questions, we would be happy to provide additional clarification.

---

> > ### Comment · Reviewer_EKuR · 2025-05-22
> >
> > This makes sense and this answers my main question!

---

### Decision · Action_Editor_RKZN · 2025-06-25

**Recommendation:** Accept as is

**Additional Comments:**

This is a solid paper; technical exposition is clear, the claims are well-rooted in the results given. Furthermore, all the reviewers were satisfied and voted to accept the paper, and I agree with them.

**Audience:**

Yes

**Audience Explanation:**

Conformal inference is an important general-purpose family of methods, and the online selection setup considered in this paper is completely plausible in practice.

**Claims And Evidence:**

Yes

**Claims Explanation:**

The claims made by the authors are clear; they highlight issues that arise in current conformal inference methods when a sequential "selection" spoils exchangeability assumptions, and they motivate, propose, and evaluate alternative methods. After the review process, all reviewers were satisfied with the clarity of this submission.